# Histone H3 serine-57 is a CHK1 substrate whose phosphorylation affects DNA repair

Nikolaos Parisis[1,2,3,4], Pablo D. Dans[5,6,12], Muhammad Jbara[7,8], Balveer Singh[9], Diane Schausi-Tiffoche[9], Diego Molina-Serrano[9], Isabelle Brun-Heath[5], Denisa Hendrychová[1,2,13], Suman Kumar Maity[7], Diana Buitrago[5], Rafael Lema[5], Thiziri Nait Achour[1,2], Simona Giunta[10,11], Michael Girardot[1], Nicolas Talarek[1], Valérie Rofidal[3], Katerina Danezi[1,2], Damien Coudreuse[9,14], Marie-Noëlle Prioleau[4], Robert Feil[1], Modesto Orozco[5], Ashraf Brik[7], Pei-Yun Jenny Wu[9,14], Liliana Krasinska[1,2,15] ✉ & Daniel Fisher[1,2,15] ✉

Histone post-translational modifications promote a chromatin environment that controls transcription, DNA replication and repair, but surprisingly few phosphorylations have been documented. We report the discovery of histone H3 serine-57 phosphorylation (H3S57ph) and show that it is implicated in different DNA repair pathways from fungi to vertebrates. We identified CHK1 as a major human H3S57 kinase, and disrupting or constitutively mimicking H3S57ph had opposing effects on rate of recovery from replication stress, 53BP1 chromatin binding, and dependency on RAD52. In fission yeast, mutation of all H3 alleles to S57A abrogated DNA repair by both non-homologous end-joining and homologous recombination, while cells with phospho-mimicking S57D alleles were partly compromised for both repair pathways, presented aberrant Rad52 foci and were strongly sensitised to replication stress. Mechanistically, H3S57ph loosens DNA-histone contacts, increasing nucleosome mobility, and interacts with H3K56. Our results suggest that dynamic phosphorylation of H3S57 is required for DNA repair and recovery from replication stress, opening avenues for investigating the role of this modification in other DNA-related processes.

DNA replication fork processivity is hindered by a variety of mechanisms including nucleotide depletion, DNA lesions or endogenously difficult to replicate sequences, collectively known as replication stress[1]. This is a hallmark of cancer cells[2] that promotes chromosomal instability[3], by leading to collapse of stalled replication forks and formation of DNA double strand breaks (DSB), toxic chromosomal lesions that must be repaired to avoid chromosome loss and maintain cell viability. DSB repair involves either non-specific joining of ends by

[1]IGMM, CNRS, INSERM, University of Montpellier, Montpellier, France. [2]Equipe labellisée Ligue contre le Cancer, Paris, France. [3]BPMP, CNRS, INRA, Montpellier SupAgro, University of Montpellier, Montpellier, France. [4]Institut Jacques Monod, CNRS, University Paris Diderot, Paris, France. [5]IRB Barcelona, BIST, Barcelona, Spain. [6]Bioinformatics Unit, Institute Pasteur of Montevideo, Montevideo, Uruguay. [7]Schulich Faculty of Chemistry, Technion Israel Institute of Technology, Haifa, Israel. [8]School of Chemistry, Raymond and Beverly Sackler Faculty of Exact Sciences, Tel Aviv University, Tel Aviv, Israel. [9]IGDR, CNRS, University of Rennes, Rennes, France. [10]The Rockefeller University, New York, NY, USA. [11]Laboratory of Genome Evolution, Department of Biology and Biotechnology "Charles Darwin", University of Rome Sapienza, Rome, Italy. [12]Present address: Department of Biological Sciences, CENUR North Riverside, University of the Republic (UdelaR), Salto, Uruguay. [13]Present address: Department of Experimental Biology, Faculty of Science, Palacký University Olomouc, Olomouc, Czech Republic. [14]Present address: IBGC, CNRS, University of Bordeaux, Bordeaux, France. [15]These authors jointly supervised this work: Liliana Krasinska, Daniel Fisher. ✉e-mail: liliana.krasinska@igmm.cnrs.fr; daniel.fisher@igmm.cnrs.fr

canonical non-homologous end-joining (NHEJ), or homologous recombination (HR) pathways that include gene conversion (GC, or "classical" HR), single strand annealing (SSA), break-induced repair (BIR) or microhomology-mediated end-joining (MMEJ). Though mechanistically distinct, these pathways nevertheless have overlapping system components[4–6]. Because GC requires a sister chromatid template, it is coupled to DNA replication[7,8] and is a high fidelity process. However, GC is slow and easily saturated in cells, and at high DSB load, SSA takes over[9]. NHEJ is the dominant DNA repair process and occurs throughout the cell cycle[7,8]; its components are also involved in replication stress responses[10]. Likewise, the HR machinery protects stalled replication forks against excessive resection and allows their restart by fork reversal or lesion bypass[11–13]. In both HR and NHEJ, chromatin is remodelled around the DSB, with eviction of nucleosomes to allow access of repair machinery[14–16]. Subsequent chromatin reformation may favour HR by promoting recruitment of the Rad51 recombinase[17].

The usage of these different DNA repair pathways is affected by histone post-translational modifications (PTMs)[18]. Ubiquitinated H2A lysine-15 (H2AK15) binds either BRCA1, which promotes end resection by the MRE11-RAD50-NBS1 (MRN) endonuclease, or 53BP1, which inhibits end-resection and favours NHEJ, in a competing manner[19]. Methylation of H4K20 also favours NHEJ by binding a second domain of 53BP1[20] and preventing binding of the BRCA1-partner BARD1[21]. PTMs of the histone core also affect DNA repair. H3K56 is located close to the DNA entry/exit site, and its acetylation (H3K56ac) promotes transcription[22] and replication stress responses[23–26]. Replication-coupled chromatin assembly involves histone chaperone-mediated incorporation of nucleosomes marked with H3K56ac. Phosphorylation of the adjacent residue, serine-57, was detected in a large-scale proteomics screen in human cells[27] and in the nematode *C. elegans*[28] but

has remained undetectable in the budding yeast *S. cerevisiae*[29], and its functions, if any, are unknown. However, by analogy with H3K56ac, phosphorylation of S57 might be expected to alter local electrostatic interactions with DNA. Genetic analysis in budding yeast revealed complex interactions between alleles encoding mimics of constitutive modification of K56 and S57 in response to agents inducing replication stress[29], including methylmethane sulphonate (MMS) and hydroxyurea. This suggests that phosphorylation of H3S57 (H3S57ph), like H3K56ac, could potentially influence DNA repair or responses to replication stress.

DNA replication is controlled by protein kinases, including cyclin-dependent kinases (CDKs) and checkpoint kinases[30,31]. In this study, we set out to discover novel chromatin phosphorylation events that promote a favourable environment for replication using *Xenopus* egg extracts, and identified H3S57ph. We studied its regulation and functions using a multidisciplinary approach, including proteomics, experiments with mammalian cultured cells, kinome-wide screening, in vitro assays, genetics in fission and budding yeast, and whole atom molecular dynamics simulations.

## Results

### Identification of H3S57 phosphorylation

To identify novel S-phase chromatin phosphorylations, we performed phosphoproteomics analysis of replicating chromatin in *Xenopus* egg extracts (XEE). This model supports synchronous DNA synthesis in vitro and has been pivotal in identifying conserved DNA replication and DNA damage-response (DDR) mechanisms[32]. We identified a histone H3 peptide phosphorylated on Ser-57 (H3S57ph, Fig. 1a, Supplementary Data 1), a conserved amino acid at the entry/exit point of DNA on the nucleosome (Supplementary Fig. 1a, b). To characterise H3S57ph, we generated a highly specific antibody by

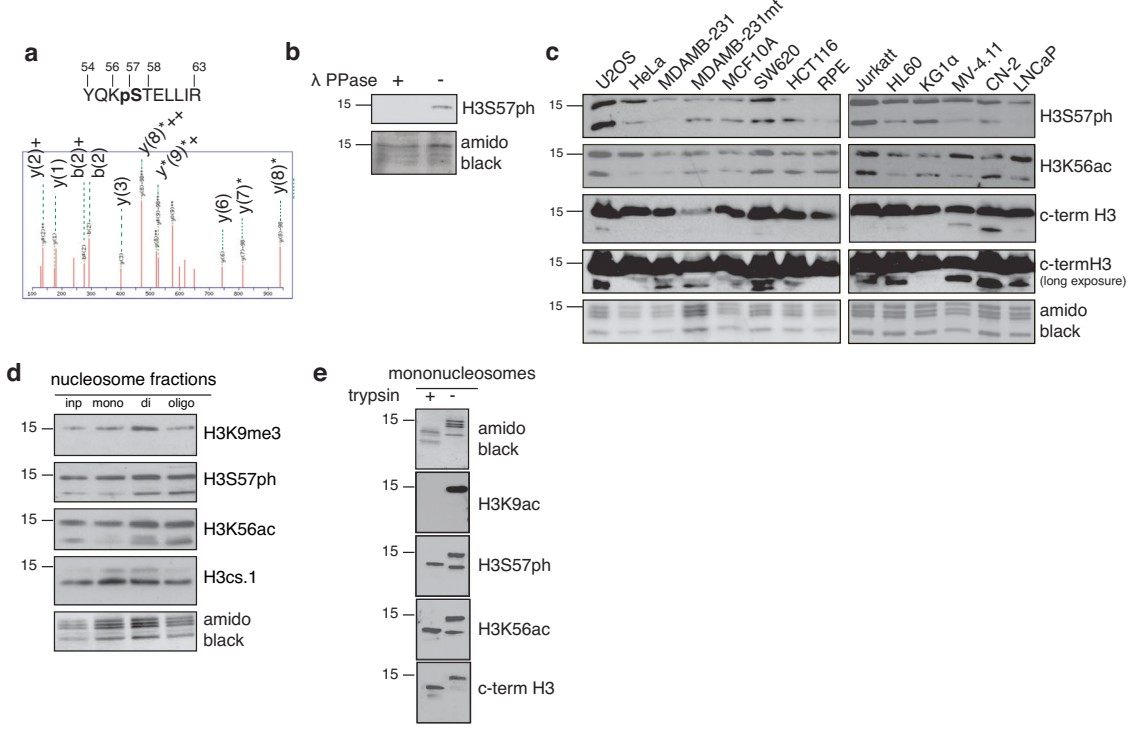

**Fig. 1 | Identification of Histone H3 Ser-57 phosphorylation (H3S57ph) in vertebrates. a** MS/MS spectrum of a histone H3 peptide bearing phosphorylated Ser-57 noted by the ions y7, y8 and y9 with neutral loss (represented by *); 5 different egg extracts were used. **b** Western blot analysis (WB) of chromatin purified from interphase *Xenopus* egg extracts during DNA replication, treated with buffer or lambda phosphatase, amidoblack for loading control; *n* = 2 independent

experiments. **c** WB (18% PAGE) of acid-extracted histones from a panel of human cell lines; *n* = 1 experiment. **d** WB (18% PAGE) of sucrose gradient-separated nucleosome fractions. H3.cs1 recognises the clipped form of H3; *n* = 2 independent experiments. **e** WB (18% PAGE) of purified, native mono-nucleosomes after in vitro cleavage of their N-terminal tails by trypsin; *n* = 3 independent experiments. Source data are provided as Source Data file.

affinity purifying the serum against the phosphorylated $H3_{51-72}$ peptide and depleting it against both the corresponding non-phosphorylated peptide and against a phosphorylated peptide surrounding serine-10, which is in a very similar sequence context. This H3S57ph antibody efficiently recognised phosphorylated $H3_{51-72}$ peptides but not phosphorylated $H3_{5-20}$ peptides, while H3K56ac did not prevent recognition of the H3S57ph epitope (Supplementary Fig. 2a). We also chemically synthesised full-length histone H3, either unphosphorylated, or with stoichiometric phosphorylation of Ser-57 (Supplementary Fig. 2b; the histone synthesis procedure was similar to that previously reported[33]). Our antibody recognised only chemically synthesised histone H3 containing S57ph (Supplementary Fig. 2c), confirming its specificity. This H3S57ph antibody detected a 15 kDa (the size of histone H3) band on immunoblots of chromatin from XEE, which was lost upon lambda phosphatase treatment (Fig. 1b). We next assessed the presence of H3S57ph in a variety of commonly-used cultured human cell lines. In acid-extracted nuclear preparations from each cell line, we detected two bands with our H3S57ph antibodies (Fig. 1c). In U2OS cells, in which H3S57ph is abundant, both bands were present in nuclear extracts, acid extracted histone preparations, or micrococcal nuclease-digested chromatin; salt extraction showed that H3S57ph is mainly present in mono-nucleosomes (Supplementary Fig. 2d). Similarly, two bands were observed with commercial H3K56ac antibodies (Fig. 1c) whose specificity we validated[34,35] (Supplementary Fig. 2e). The levels of each band varied between different cancer cell lines, while we only detected low levels of H3S57ph in non-transformed retinal pigment epithelial (RPE) cells (Fig. 1c). This variability in levels of H3S57ph suggests that the mark is not constitutively present on all nucleosomes in most cell lines, and is elevated in cancer cells, which have higher levels of replication stress[2].

We wondered whether the faster migrating of the two H3S57ph bands (around 13 kDa) might correspond to tailless H3, which has been previously observed during embryonic stem cell differentiation and is thought to regulate transcription by altering chromatin modifications[36–38]. We fractionated acid-extracted preparations of U2OS cells by HPLC[39] and confirmed by immunoblotting that both bands identified by H3S57ph antibodies are H3 species (Supplementary Fig. 2f). In agreement with this conclusion, H3 c-terminus, H3K56ac, and H3cs.1 tail-cleavage specific[36] antibodies also identified the faster-migrating band, while antibodies against modifications of the histone N-terminus did not (Fig. 1d), suggesting that the lower band has lost the N-terminal tail. In support of this hypothesis, upon controlled in vitro trypsin cleavage of histone extracts[40,41], only a single, faster migrating species remained (Fig. 1e). Collectively, these data indicate that our antibody specifically detects H3S57ph as a PTM of both full-length and tailless H3.

We wondered whether H3S57ph is a cell cycle-regulated mark. We observed numerous nuclear foci of H3S57ph by immuno-fluorescence (Fig. 2a), whose intensity increased as cells progress into G2 (Fig. 2b), consistent with increasing H3 abundance during the cell cycle. To test this, we synchronised cells in mitosis with noco-dazole or in early S-phase with thymidine, and released them from the blocks (Fig. 2c). Cells blocked in mitosis with nocodazole had high H3S10ph, as expected, but low H3S57ph levels (Fig. 2d). This confirms that our antibody does not cross-react with H3S10ph and shows that H3S57ph levels are low in mitosis. In contrast, cells arrested in early S-phase by double thymidine block showed abundant H3S57ph, which decreased upon release of the block before increasing again upon replication (Fig. 2d). We observed similar behaviour in cycling mouse embryonic stem cells (Supplementary Fig. 3a). These data suggested that H3S57ph might be incorporated

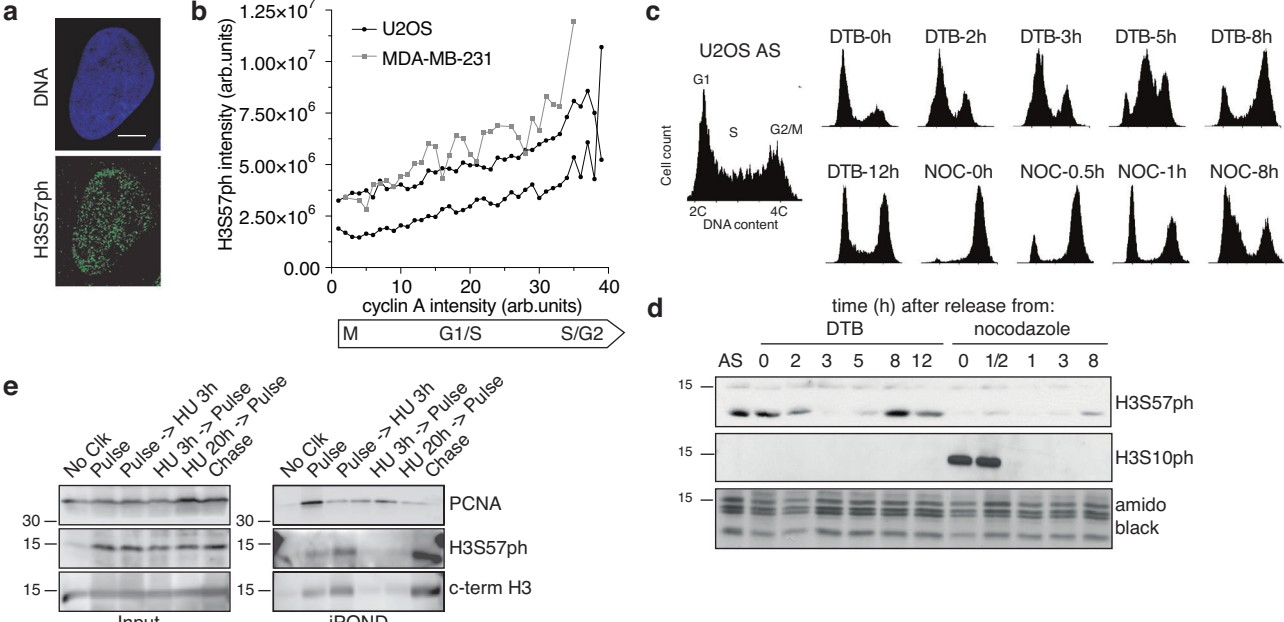

**Fig. 2 | H3S57ph levels are regulated during the cell cycle. a** Representative immunofluorescence image of a U2OS cell stained with DAPI (DNA) and H3S57ph antibody; *n* > 10 independent experiments, over 100 cells were analysed each time. Scale bar, 10 μm. **b** Plotting of H3S57ph and cyclin A signal intensity quantified from immunofluorescence images. Cyclin A staining was used as a cell cycle phase marker (mean ± SD of 1102 and 1247 U2OS cells, and 362 MDA-MB-231 cells; U2OS, *n* = 2, MDA-MB-231, *n* = 1 experiment). **c** FACS profiles of U2OS cells, asynchronously growing (AS) or synchronised with either double thymidine block (DTB) or nocodazole (NOC), and released for the indicated time; *n* > 5 independent

experiments; 10,000 cells an analysed. The gating strategy is presented in Supplementary Fig. 11. **d** Western blot analysis of cells from the synchronisation experiment in **c**. H3S10ph was used as mitotic marker and to confirm non-cross reactivity with anti-H3S57ph antibody; amidoblack was used for loading control; *n* = 2 independent experiments. **e** WB of proteins present at replication forks, purified by iPOND. Pulse, 10 min EdU or 30 min upon HU; Chase, 1 h thymidine upon pulse; HU, 5 mM for 3 h or 0.2 M for 20 h; No Clk, sample without click reaction for negative control; *n* = 2 independent experiments. Source data are provided as Source Data file.

into newly formed chromatin after the passage of replication forks and that it is elevated upon replication arrest. By isolation of proteins on nascent DNA (iPOND[42]), we confirmed that this is the case (Fig. 2e). However, unlike PCNA, which is specific for replication forks and is reduced on iPOND after thymidine chase, H3S57ph was further increased. Thus, in contrast to H3K56ac in yeast[23,26], H3S57ph is not transiently present around replication forks. Genome-wide ChIP-seq of H3S57ph in exponentially growing or hydroxyurea-arrested U2OS cells did not reveal a specific genomic localisation (Supplementary Fig. 3b, c), suggesting that it is ubiquitous or independent of sequence.

### CHK1 kinase is the major H3S57 kinase

Next, we sought to identify H3S57ph "writers". We screened for candidates by performing in vitro assays with 190 Ser/Thr human kinases and the H3$_{51-72}$ peptide as substrate. Of the four kinases with high activity against this substrate, the DNA damage response kinase CHK1 is the only constitutively expressed nuclear kinase (Fig. 3a, and Supplementary Data 2). To determine whether CHK1 is a major H3S57 kinase, we evaluated three criteria: its ability to directly phosphorylate H3S57, its presence on chromatin at the time when H3S57 is phosphorylated, and the dependency of H3S57ph on this kinase. First, as assayed by quantification of $^{33}$P radioactive phosphate incorporation,

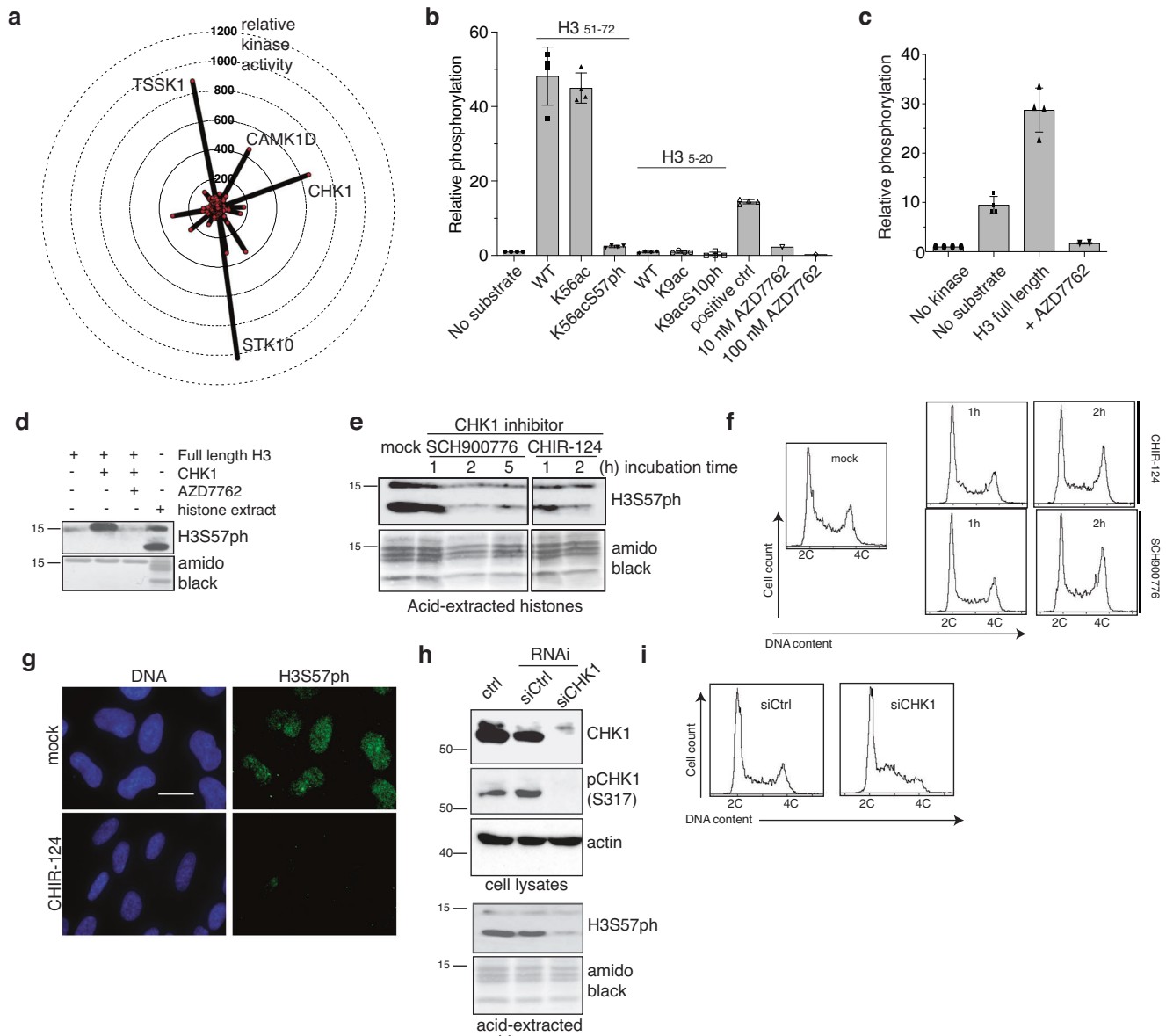

**Fig. 3 | Checkpoint kinase 1 (CHK1) is the H3S57ph kinase in human cells.**
**a** Radar plot with activity values of 190 kinases in kinase assays with H3$_{51-72}$ peptide as substrate ($n = 1$ experiment). Only kinases above the threshold are labelled, see Supplementary Data 1 for full results. In vitro kinase assays using $\gamma^{33}$P-ATP, recombinant CHK1 and H3 peptides, with the indicated modifications (**b**; $n = 4$) or full-length H3 (**c**; $n = 4-5$) as substrates; error bars, mean ± SD; results are normalised to, and represented as fold increase over, the control assays without kinase. **d** In vitro kinase assays as in **c** but with non-radioactive ATP and analysed by western blotting; $n = 1$ experiment. The histone extract was used as positive control for the WB. U2OS cells were treated with CHK1 inhibitors (SCH900776, 5 μM; CHIR-124, 2 μM) for indicated times, and analysed by WB of acid-extracted histones (**e**), FACS cell cycle profiles (**f**), or by immunofluorescence of fixed cells (**g**); $n = 2$ independent experiments. All lanes come from the same membrane/exposure, unrelated lanes were removed. DNA was stained with DAPI; scale bar, 20 μm. Knockdown of CHK1 by RNA interference in U2OS cells, analysed subsequently by WB (**h**) of total cell lysates (top) and acid-extracted histones (bottom), and FACS cell cycle profiles (**i**); amidoblack staining was used for loading control; siCtrl, siRNA targeting the firefly luciferase gene; $n = 2$ independent experiments. Source data are provided as Source Data file.

purified recombinant CHK1 efficiently phosphorylated H3S57 peptides in vitro irrespective of K56 acetylation (Fig. 3b), but not a similar peptide already phosphorylated on S57 (Fig. 3b). This indicates that S57 is the only CHK1 site on this peptide, i.e. threonine-58 is not a target. In addition, as assessed by the appearance of the epitope for our H3S57ph antibody, CHK1 could also phosphorylate full-length recombinant histone H3 on serine-57 (Fig. 3c, d). Although an in vitro study reported that DYRK1A may phosphorylate H3S57[43], we found that purified DYRK1A could not efficiently phosphorylate either the H3$_{51-72}$ peptide, or Ser-57 on full length recombinant H3, although it could phosphorylate full length H3 at Thr-45, as previously reported[44] (Supplementary Fig. 4a, b). Second, we confirmed by immunoblotting that CHK1 is localised on chromatin during a double thymidine block and release (Supplementary Fig. 4c), thus, when H3S57ph is also present. Third, to determine whether H3S57ph is CHK1-dependent, we depleted CHK1 by siRNA-mediated silencing, or inhibited its kinase activity by treatment of cells with two different specific CHK1 inhibitors, CHIR-124 and SCH900776. In each case, these treatments strongly reduced H3S57ph levels (Fig. 3e–i). The loss of H3S57ph after short CHK1 inhibitor treatments (Fig. 3e) occurred in the absence of S-phase arrest (Fig. 3f), implying that S57ph is a dynamic modification. Longer term, loss of CHK1 by knockdown in U2OS cells increased the S-phase fraction (Fig. 3h, i), while its inhibition prevented completion of DNA replication in *XEE* (Supplementary Fig. 4d) and in synchronised U2OS cells (Supplementary Fig. 4e). This is consistent with the findings that the *Chk1* gene is essential for viability in mice and promotes efficient S-phase progression in mouse embryonic fibroblasts and chicken DT40 cells[45,46]. Collectively, these results establish CHK1 as the most likely candidate for the H3S57 kinase in human cells.

## Mutations of H3S57 affect responses to replication stress

Since CHK1 is a DNA damage-responsive kinase, we asked whether H3S57ph levels are affected by treatments that cause replication stress. These include conditions that induce single-stranded DNA, allowing the formation of G-quadruplex structures (G4)[47], whose high thermodynamic stability hinders the replication machinery. G4 can be detected by the BG4 antibody[48], and we reasoned that this should serve as a good marker for replication stress. 24 hours of G4 stabilisation by pyridostatin[49] (PDS), like treatments with hydroxyurea (HU), etoposide or ICRF-193, stimulated BG4 foci formation and led to the appearance of γH2A.X foci, a marker of DNA damage[50] (Supplementary Fig. 5a). However, although H3S57ph staining was moderately increased by treatments that arrest cells in S-phase or G2 (Supplementary Fig. 5b), this is consistent with the increased overall level of H3 that occurs due to histone synthesis in S-phase (Fig. 2b). We then asked whether H3S57ph localises to sites of DNA damage. We thus subjected cells to ionising radiation (IR – 10 Gy), which induces double-stranded DNA breaks (DSBs). In U2OS cells, IR resulted in reorganisation of H3S57ph into a few discrete foci, some of which colocalised with γH2A.X (Supplementary Fig. 5c). Furthermore, immunoprecipitation of H3S57ph from purified mononucleosomes[51] showed that H3S57ph and γH2A.X may be present on the same nucleosome (Supplementary Fig. 5d). We conclude that H3S57ph is not a general marker of DSBs but might be involved in DNA repair or responses to replication stress.

To examine this possibility, we analysed the effects of disruption of H3S57ph. Replacement of all 16 genes encoding histone H3 in human cells on both alleles is technically extremely challenging, and we failed to derive stable lines expressing multiple shRNA targeting all H3 genes. However, we reasoned that since excess histone expression hinders S-phase progression in both yeast and human cells[52], overexpressing WT or H3S57 mutants should allow us to evaluate the role of the modification of this residue in the replication stress response. We thus stably transfected U2OS cells with a plasmid expressing histone H3.1-FLAG WT or alleles mutated at H3S57 to non-phosphorylatable (S57A) or phospho-mimic (S57D) amino acids. In U2OS clones stably expressing these alleles at similar levels (Supplementary Fig. 6a), WT H3 caused a slight decrease in cell proliferation, as expected (Fig. 4a). H3S57D cells had a similar but exacerbated phenotype, while, in contrast, H3S57A accelerated cell proliferation (Fig. 4a). H3S57A expression also quickened progression through S-phase after release from an HU-mediated arrest (Fig. 4b). This suggests that phosphorylation of H3S57 might promote slower pathways of responses to replication stress, while overexpression of histones constitutively mimicking H3S57 phosphorylation appears to be toxic. To extend these analyses, we also ascertained DNA damage responses at a single cell level by flow cytometry measurement of γH2A.X staining. As expected, overexpression of WT H3 resulted in spontaneous DNA damage, with 8% of cells positive for γH2A.X. H3S57D showed a similar phenotype, implying that H3S57D does not hinder cell proliferation by further increasing DNA damage (Fig. 4c). In contrast, cells overexpressing H3S57A did not present increased γH2A.X. HU treatment caused prominent γH2A.X staining in around half of cells irrespective of H3 overexpression. Exogenous WT H3 or H3 S57D slowed the recovery after removal of HU, while in S57A mutant cells, the recovery was accelerated (Fig. 4c). These results therefore suggest that S57 phosphorylation might differentially affect DNA repair pathways.

Recovery of stalled replication forks generally involves the HR pathway, which implicates resection of a single-ended DNA followed by strand invasion. Resection is counteracted by binding of 53BP1, which may protect against over-resection at stalled replication forks[10]. Resection reveals single-stranded DNA (ssDNA) which is bound by RPA prior to strand invasion and repair. To assess the possibility that H3S57ph might affect resection, we treated cells overexpressing WT H3, H3S57A and H3S57D with HU, and analysed ssDNA levels in single cells by a native BrdU FACS assay. Fewer HU-treated cells overexpressing WT H3 or H3S57D showed ssDNA than cells without exogenous histone expression, despite similar levels of DNA damage (Fig. 4d), while the fraction of H3S57A-expressing cells with ssDNA was close to controls (Fig. 4d). Similar results were obtained by analysis of ssDNA-bound RPA, which greatly increases upon HU treatment (Supplementary Fig. 6b, c). Altogether, our results indicate that S57 phosphorylation is not required for, and on the contrary, may limit, end resection.

One possibility is that H3S57ph is involved in the choice of DNA repair pathways. Based on our observations, H3S57ph might promote DNA repair by HR (from the sister chromatid) or SSA (from the same chromatid); in the absence of phosphorylation, cells might recover by faster pathways such as fork reversal followed by branch migration, or by 53BP1 and RIF1-mediated fast fork restart[53,54]. To characterise the pathways involved in repair in cells expressing different H3 alleles, we assessed the involvement of 53BP1, which inhibits HR, and RAD52, which is dispensable for HR but promotes SSA. Compared to cells expressing WT H3, far more cells expressing H3S57A showed high 53BP1 levels upon HU treatment; in contrast, fewer cells with H3S57D showed high 53BP1 staining (Fig. 4e; note that all cells expressing exogenous H3, WT and mutants, had higher basal levels of 53BP1). Combined with the observation that cells lacking H3S57 phosphorylation have more ssDNA, this suggests that 53BP1 may protect against over-resection in these cells. We next knocked down RAD52 by siRNA and assessed recovery from replication stress. We observed that loss of RAD52 triggered spontaneous DNA damage in H3S57A-expressing cells and led to higher levels of DNA damage after HU treatment (Fig. 4f). Furthermore, while RAD52 knockdown had little effect on HU recovery in cells without exogenous histone expression or in cells expressing H3S57A, the recovery was accelerated in cells expressing WT H3 or H3S57D (Fig. 4f). This suggests that S57 phosphorylation promotes slow RAD52-dependent repair, likely SSA, while reducing this modification stimulates repair by a faster 53BP1-dependent mechanism.

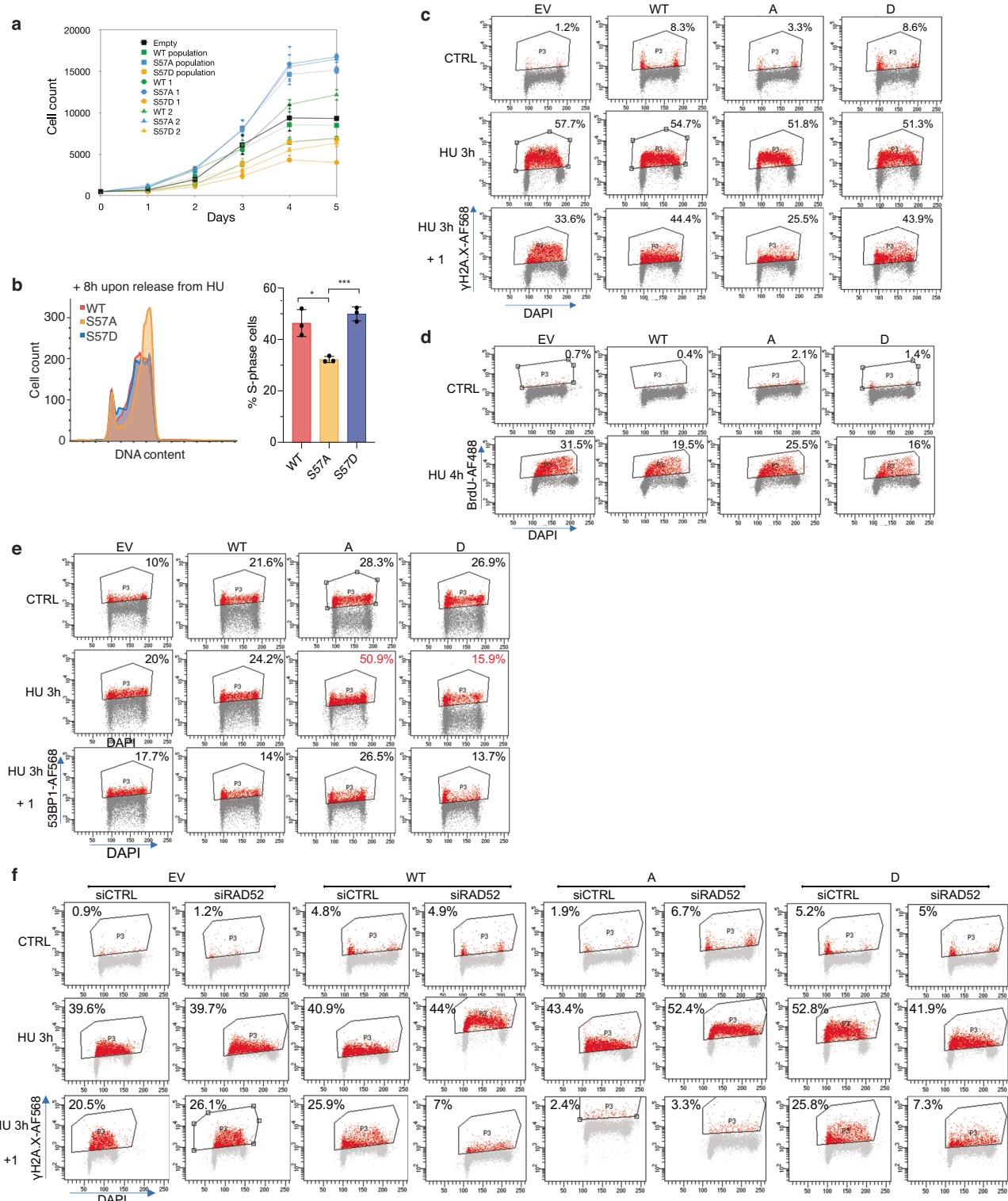

The experiments described above give clues to the cellular functions of H3S57ph, but may reflect partial or compound phenotypes, as they involve overexpression of altered histones, which compete with endogenous proteins and also potentially induce replication stress. They are therefore insufficient for drawing definitive conclusions about the pathways involved in replication stress responses that depend on H3S57ph. They also do not allow determination of whether or not H3S57ph is an essential mark; eliminating CHK1, the kinase that mediates H3S57ph in human cells causes S-phase arrest, but this is likely due to substrates other than H3S57.

Since budding yeast *Saccharomyces cerevisiae* expressing only alleles of H3 in which K56 or S57 are mutated show phenotypic alterations upon treatment with genotoxic agents[29], we first performed histone extractions from wild-type budding yeast cells, or cells in which both genes encoding histone H3 were deleted and a H3S57A variant was ectopically expressed[29]. As with human cells, two bands recognised by our antibody were present in WT histone preparations in exponentially growing yeast, or yeast treated with MMS (Supplementary Fig. 7a), a DNA alkylating agent that induces replication stress[55], but no signal was detected in cells only expressing H3S57A alleles

**Fig. 4 | Cells overexpressing H3S57 mutants use different pathways of replication stress recovery. a** U2OS cells expressing an empty vector or FLAG-tagged H3.1wt, S57A or S57D constructs under the CMV promoter, in initial populations and two isolated clones, were plated at low density and monitored for proliferation during 5 days. Experiments were performed in triplicates and results are shown as mean ± SD. **b** Left, overlay image of FACS profiles of cells collected 8 h after release from a 20 h HU block. Right, bar graph representing the percentage of S-phase cells at the HU + 8 h timepoint. *$p = 0.01$, ***$p = 0.0005$ (Student's $T$ test); error bars, mean ± SD; $n = 3$ independent experiments. The gating strategy is presented in Supplementary Fig. 11. **c** Analysis of γH2A.X by FACS in non-treated cells, or cells treated with HU for 3 h, and allowed to recover for 1 h; DAPI was used to counterstain DNA; the fluorochrome used is indicated (AF, AlexaFluor). Doublets and debris were excluded, and 10 000 cells were analysed per sample. Gates were arbitrarily set in EV control conditions and implemented on the other conditions. The gating strategy is provided in Supplementary Fig. 11. Representative images from 3 independent experiments are shown. **d** As in **c**, but ssDNA was assessed by analysis of native BrdU incorporation in native conditions, in either control cells or cells exposed to HU for 4 h ($n = 2$ independent experiments). **e** As in **c**, but 53BP1 was analysed ($n = 2$ independent experiments). **f** As in **c**, but γH2A.X was analysed in cells previously submitted to two rounds (48 h) of siRNA treatment to down-regulate RAD52. Final gates were repositioned manually where necessary due to drift; $n = 1$ experiment. Source data are provided as Source Data file.

(Supplementary Fig. 7a). These data indicate that H3S57ph is a conserved histone PTM.

To investigate the functions of H3S57 in vivo, we next turned to the fission yeast *Schizosaccharomyces pombe*, whose pathways for cell cycle regulation, replication stress responses, and DNA repair are closely conserved with those in mammals. We replaced all three genes encoding fission yeast histone H3 with alleles that express H3S57A or H3S57D at the endogenous histone H3 loci (Supplementary Fig. 7b). As with budding yeast, our H3S57ph antibodies detected a clear signal in extracts of WT fission yeast cells in a phosphorylation-dependent manner, but only background staining was observed in extracts of cells that express only H3S57A (Fig. 5a, b). Both H3S57A and H3S57D mutant fission yeast strains were viable at 32 °C and showed no significant proliferation defects (Supplementary Fig. 7c), although they both displayed temperature sensitivity phenotypes (Supplementary Fig. 7d). These results demonstrate that H3S57ph is not essential in normal growth conditions but may be crucial for cellular function in challenging situations.

We then asked whether the role for H3S57ph in the replication stress response described above in human cells is conserved in fission yeast. Importantly, H3S57A and H3S57D cells were sensitive to MMS, camptothecin, and HU (Fig. 5c), further supporting a role for H3S57ph in the response to replication stress. In particular, the H3S57D mutant was acutely sensitive to these challenges, with phenotypes that are similar to cells lacking the checkpoint kinase Chk1 (the fission yeast CHK2 orthologue; the fission yeast homologue of CHK1 is Cds1; Fig. 5c). In contrast, these strains were not sensitive to UV irradiation, indicating that they do not have a general defect in DNA repair. Building on these observations, we evaluated the interplay between H3S57 phosphorylation and the replication stress response. First, we assessed the formation of Rad52 foci, a marker for spontaneous and induced DNA damage, in our different genetic backgrounds. We found that fission yeast harbouring H3S57D displayed a ~5-fold increase in the number of cells with either >2 or large, aberrant Rad52 foci in the absence of replication stress-inducing agents, with H3S57A showing a more modest defect (Fig. 5d). These results raise the possibility that deregulation of S57 phosphorylation, in particular a constitutive mimicking of the modification, brings about DNA damage even in non-challenging conditions. This idea is supported by the modest increase in cell length at division of H3S57D mutants, which may be due to pathways that delay mitosis until DNA repair is completed (Supplementary Fig. 7b). Moreover, an elevated level of DNA damage in these cells, when combined with replication stress inducers, might underlie the sensitivity of the H3S57D mutant to these drugs (Fig. 5c). We then asked whether alteration of S57 might affect the rates of recovery from replication stress, as observed for human cells overexpressing histone H3 mutants. To this end, we monitored the behaviour of cells treated with HU and then allowed to re-enter the cell cycle upon HU removal. Both the cell cycle arrest and the recovery from an HU block proceeded with similar kinetics for WT, H3S57A, and H3S57D, suggesting that the process of replication restart is not strongly dependent on H3S57 phosphorylation in fission yeast (Supplementary Fig. 7e).

Next, we explored the effects of H3S57 modification on DNA repair pathways, using two different plasmid-based assays (Supplementary Fig. 8). The first assesses NHEJ using an autonomously replicating vector that is cleaved within sequences that lack homology to the yeast genome. Its repair by NHEJ results in vector re-ligation, which allows cells to grow on selective medium. The second assay evaluates HR-dependent repair pathways using a non-replicating plasmid that is linearised within a fission yeast gene; its genomic integration by HR permits growth of cells on selective medium. In both assays, cells were co-transformed with a plasmid conferring resistance to nourseothricin to normalise for transformation efficiency. We verified the specificity of these assays by comparing WT cells with cells lacking DNA ligase 4 (*lig4*), which is required for NHEJ, or Rad52, which is necessary for HR (Fig. 5e). Our results showed that H3S57D significantly compromised both NHEJ and HR (Fig. 5e). Surprisingly, given their relatively moderate sensitivity to replication stress inducers, H3S57A mutants almost completely abolished repair using either NHEJ or HR (Fig. 5e). These findings thus uncover a key role for H3S57ph in both NHEJ and HR in fission yeast and highlight a potential importance for the dynamics of this modification in DNA repair.

## H3S57 phosphorylation loosens DNA-nucleosome interactions

Structural biology of histone PTMs on the core and entry/exit regions highlights their key role in controlling nucleosome thermodynamic stability, which is critical for accessing DNA[56]. To assess the possibility that H3S57ph would alter nucleosomal dynamics, we performed all atom molecular dynamics (MD) simulations of the nucleosome[57], either phosphorylated or not on H3S57. We found that both nucleosomes with or without the phosphorylation were stable and displayed a RMSD (Root Mean Squared Deviation) from the common initial structure of 5.1 Å and 2.4 Å, respectively, with mobility concentrated in the intrinsically disordered tail of H3 in both cases (Supplementary Fig. 9a, b, Fig. 6a). The higher RMSD of the H3S57ph nucleosome is mainly due to a conformational change occurring at the N-terminus, where two new short helices are formed (Fig. 6a, Supplementary Fig. 9c, d), while there are no perceptible structural distortions on the nucleosome protein core in the sub-microsecond timescale. We also observed a systematically higher RMSD value for the DNA residues (133–163) at the entry/exit point of the nucleosome, that are in contact with the H3 tail (labelled herein as DNAe/e, Fig. 6b, c). The DNAe/e in H3S57ph was slightly but systematically displaced with respect to the WT (average RMSD difference of 0.5 Å) due to an altered network of local atomic contacts. Analysing these interactions along the trajectory for the WT, we found that H3S57 forms a hydrogen bond (Hbond) with H4R40 for 99% of the time, while K56 interacts with the phosphate group of dC156 of the DNAe/e (57%), and with H2AK128 (35%). Upon H3S57 phosphorylation, the H3S57 bond to H4R40 is replaced by H3S57ph bonds with H3R53 (57%) and H3K56 (32%) (Fig. 6c), while K56-dC156 and R53-dT155 interactions on the DNAe/e backbone are reduced, from 57% to 23% and 73% to 37%, respectively (Fig. 6c). Thus, two H3-DNA H-bond stabilising interactions are lost, while S57ph partially sequesters R53 and K56, thus reducing the electrostatic

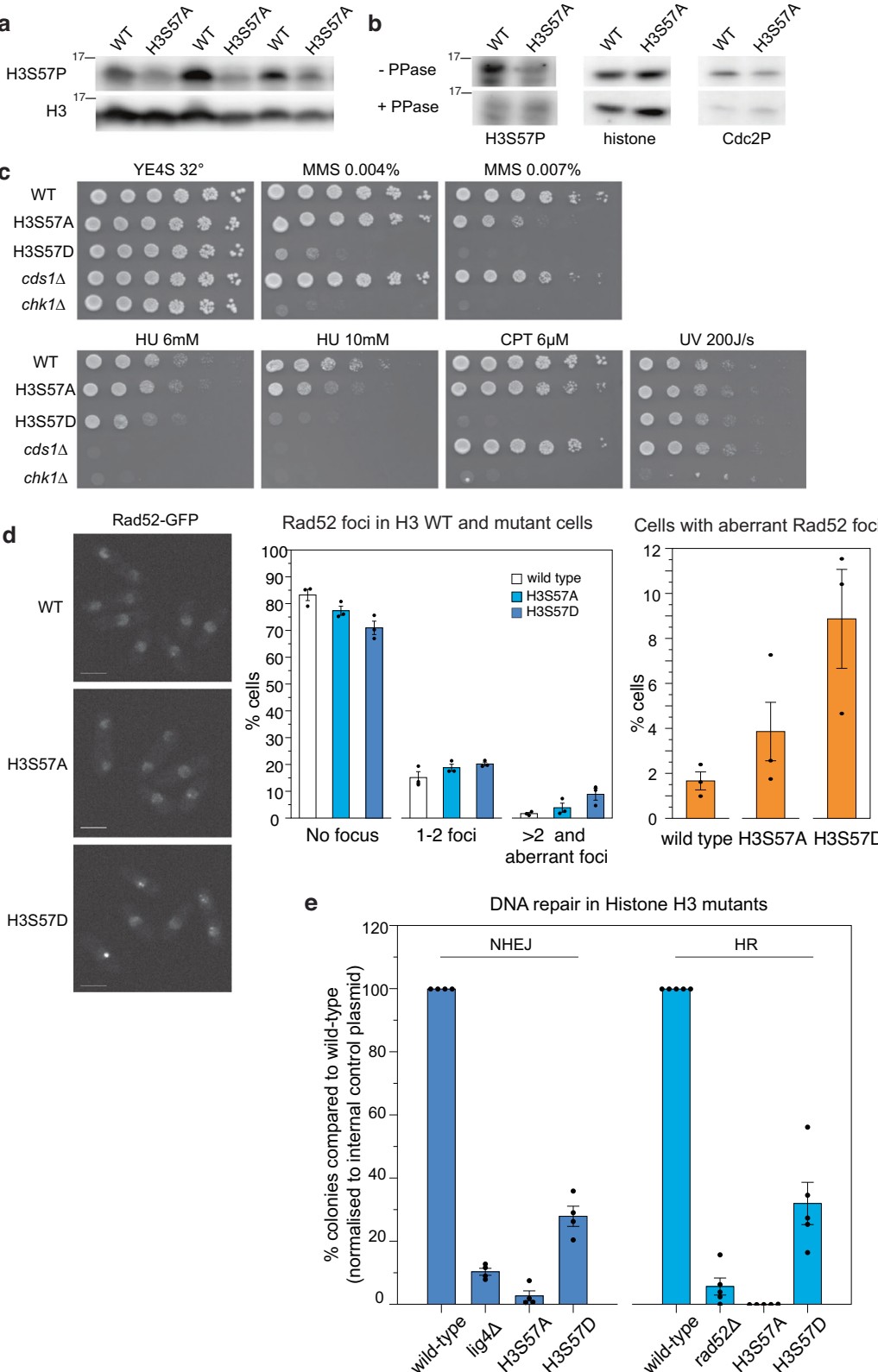

attraction between histone H3 and the DNA backbone in the entry/exit point (Supplementary Fig. 9e). This should lead to more accessible chromatin and is likely to hinder K56 acetylation.

To confirm the interaction between H3K56 and H3S57ph experimentally, we first assessed whether S57 phosphorylation affects the acetylation of K56. We thus performed in vitro acetylation assays using synthetic histone H3 proteins in which S57 is

phosphorylated or not. Indeed, as predicted, acetylation of K56 by CPB was strongly reduced by H3S57ph (Fig. 6d). To determine whether S57ph increases nucleosome mobility in vivo, we performed micrococcal-nuclease digestion of chromatin coupled with high-throughput sequencing (MNase-seq), from budding yeast expressing only H3S57E or H3S57A alleles, respectively. While nucleosome positioning was globally unaffected (Supplementary Table 1), there

**Fig. 5 | H3S57 is phosphorylated and regulates DNA damage repair pathway choice in fission yeast. a** Whole cell extracts from the indicated strains were analysed by Western blot to detect H3S57ph and histone H3. Each wild-type (WT) and H3S57A pair represents a set of independent biological repeats ($n = 3$). **b** Lambda phosphatase treatment results in loss of detection of H3S57ph by a modification-specific antibody. Dephosphorylation does not alter overall H3 levels. Phosphorylation of the Cdc2 cyclin-dependent kinase is used as a control for the efficiency of treatment. A representative experiment is shown ($n = 3$). **c** Cells with the indicated genotypes were grown to exponential phase in YE4S and spotted on plates at 10-fold dilutions. Images were taken on the following days after spotting: day 2 for UV 200 J/s; day 3 for YES 32 °C, MMS 0.007%, MMS 0.004%, CPT 6 µM, HU 6 mM; and day 6 for HU 10 mM. **d** Live-cell imaging of Rad52-GFP in exponentially growing cells with the indicated genotypes. Scale bar: 5 µm. Graphs show the

percentage of cells with no Rad52 focus, 1-2 foci, and aberrant/3 or more foci per nucleus. Averages (bars) and standard errors (whiskers) of three independent experiments (dots) are shown, $n > 200$ cells were counted for each experiment. **e** Analysis of the repair of double-stranded DNA breaks via non-homologous end joining (NHEJ) or homologous recombination (HR) in wild-type, H3S57A, and H3S57D strains. The re-ligation (NHEJ) or genomic integration (HR) of a linearised plasmid was assessed. A control vector was co-transformed for internal normalisation. The efficiency of DNA repair for each experiment was normalised to the wild-type sample. For comparison, strains defective for NHEJ (*lig4Δ*) and HR (*rad52Δ*) were included. Graphs show averages (bars) and standard error (whiskers). Dots indicate independent experiments; $n = 4$ for NHEJ and $n = 5$ for HR. Source data are provided as Source Data file.

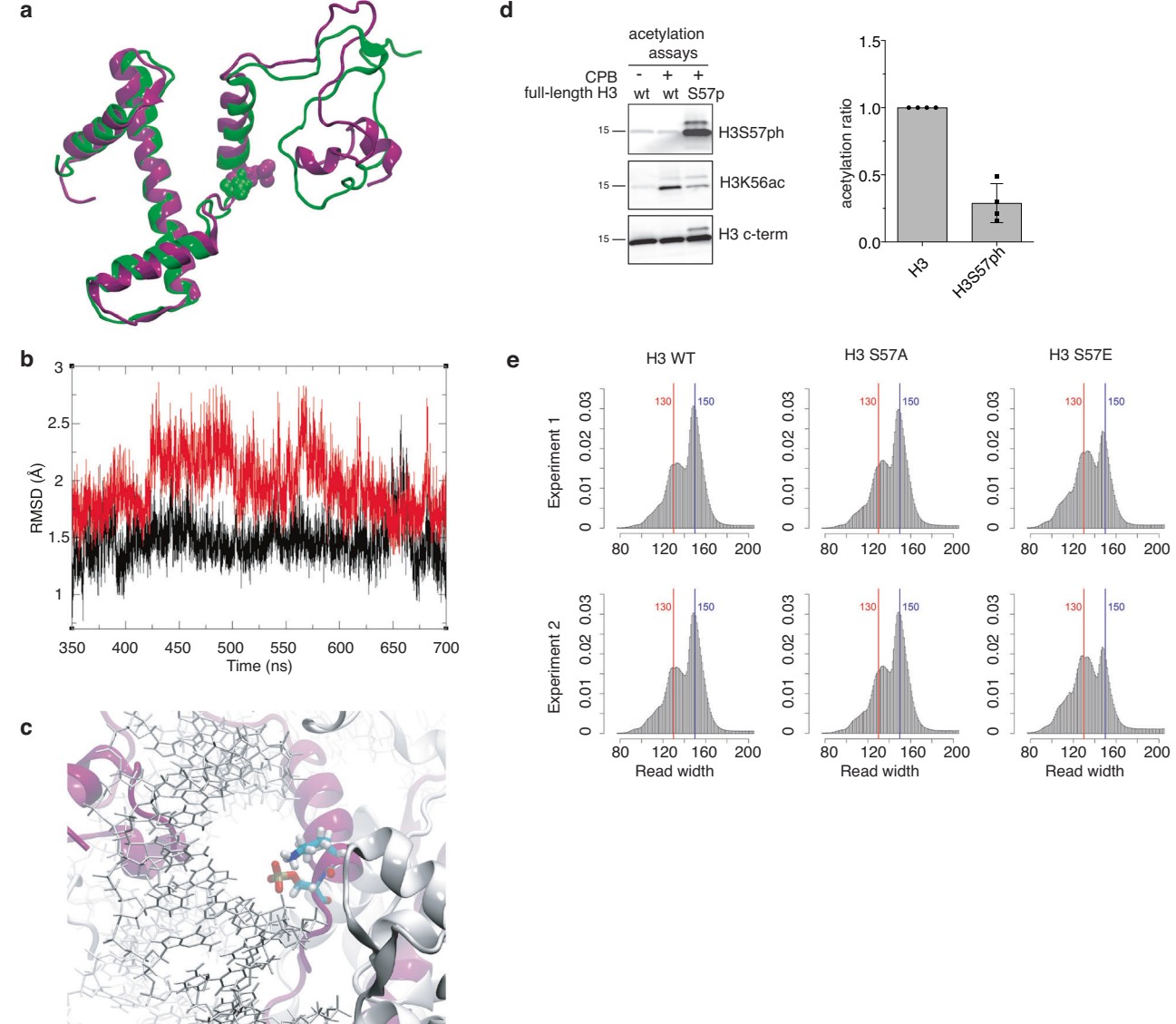

**Fig. 6 | H3S57ph increases nucleosome dynamics and interacts with H3K56. a** Comparison of final structures of molecular dynamics (MD) simulation of unphosphorylated H3 (green), and H3S57ph (violet). **b** RMSD along time of residues 133–162 (DNA in the entry/exit point of the nucleosome and in contact with H3), for unphosphorylated H3 (black) and H3S57ph (red). **c** Representative simulation snapshot showing the intra-molecular K56-S57ph interaction. **d** Left, western blot analysis of in vitro acetylation assays using CPB and synthetic full-length H3,

unphosphorylated or phosphorylated on S57 (H3S57ph). Right, bar graph with the quantification of the images of 4 independent WB experiments, representing the percentage of H3K56 acetylation between the two constructs; error bars, mean ± SD; $p < 0.0001$; $n = 4$ independent experiments. **e** Distribution of the sequenced MNase-digested fragments of around 135 bp and around 147 bp for budding yeast strains expressing WT, H3S57A and H3S57E mutants. Source data are provided as Source Data file.

was a strong increase in the amount of sub-nucleosomal fragments (<147 bp) in the H3S57E mutant compared to the WT and the H3S57A mutant (Fig. 6e, Supplementary Fig. 9f). In the S57E mutant, the average size of di-and tri-nucleosomes was 272 bp and 402 bp, respectively, compared to 292 bp and 425 bp in the WT and S57A mutants. The accumulation of nucleosomal fragments 20 bp smaller than normal suggests that the phospho-mimicking S57E mutation loosens nucleosome-DNA interactions, rendering the DNA at the entry/exit point more accessible to the micrococcal nuclease.

Loosening nucleosome-DNA interactions may well facilitate access of the DNA repair machinery, but the choice of distinct DSB repair pathways involves direct interactions between chromatin marks and specific DNA repair proteins. Hence, we also investigated the possibility that H3S57ph might provide a binding site for proteins involved in DNA repair. To this end, we performed pull-downs from nuclear extracts of asynchronously growing U2OS cells of proteins that bound to biotin-labelled $H3_{51-72}$ peptides (wild-type sequence, either unmodified or bearing S57p, or a scrambled peptide) and analysed them by mass spectrometry[58]. This identified 80 proteins that bound to H3S57ph-containing peptides but not to the beads alone, the unmodified or scrambled peptides (Supplementary Fig. 10a, b; Supplementary Data 3). Many of these were previously found at HU-stalled replication forks[59]. These include proteins with known roles in DNA repair and homologous recombination (RAD50 and RPA70)[60,61], the histone variants H2A.Z[62] and H2A.J[63], DNA polymerase θ[4], and the RuvB-like proteins reptin and pontin[64,65]. Despite not being detected in our negative controls, many of these proteins, with the exception of DNA polymerase θ, have also been found in the CRAPome of proteins frequently identified in negative controls by proteomics[66]. We validated specific candidates: the interaction of H3S57ph with pontin was confirmed by immunoblotting peptide pull-downs (Supplementary Fig. 10c), and immunoblotting of purified FLAG-tagged WT, S57A and S57D-bound proteins from U2OS cells revealed that RAD50 and RPA70 bound to WT and S57D, but much less to S57A (Supplementary Fig. 10d). Further studies will be required to assess the specificity and functional relevance of these interactions. Collectively, our data suggest that dynamic phosphorylation of H3S57 promotes DNA replication stress recovery and DSB DNA repair at an early step, by directly loosening chromatin, reducing K56 acetylation, and potentially also by binding proteins involved in DNA repair.

## Discussion

In this multidisciplinary study, we identify H3S57ph as an uncharacterised PTM of a core histone on the globular domain. We show that it occurs on both full-length and clipped histones and is incorporated into replicating chromatin. We also demonstrate that it is conserved throughout eukaryotes but is non-essential for viability. Importantly, our results uncover a conserved role for H3S57ph in replication stress responses and in DSB repair. Furthermore, we identify CHK1 as the major human kinase "writer" of H3S57ph, and provide a likely molecular mechanism of action for H3S57ph, whereby it loosens nucleosome-DNA contacts and sequesters the side-chain of lysine-56, hindering acetylation of the latter.

While a previous study reported putative roles for H3S57ph in transcriptional control via exclusion of HP1[43], the methodology relied entirely on an antibody generated against recombinant H3S57ph peptides, whose specificity remains unclear. Staining of cells with this antibody was elevated in mitosis, we thus suspect that it was not completely specific for H3S57ph, and might also recognise the mitotic mark H3S10ph, which is situated in a remarkably similar sequence context. H3S57ph was not identified by mass spectrometry, and evidence for the role of DYRK1A as a H3S57 kinase was obtained by blotting $H3_{25-66}$ peptides, which might recognise a minor population, rather than by radioactive nucleotide incorporation. Although chromatin immunoprecipitation (ChIP) of DYRK1A revealed binding to a

subset of NFκB-regulated genes, we did not confirm any specific localisation of H3S57ph by ChIP-sequencing.

In human cells, we found that exogenous expression of H3S57A reduces the replication stress caused by histone overexpression, while upon induction of replication stress, it causes accumulation of 53BP1 and promotes faster, RAD52-independent repair. Conversely, mimicking constitutive phosphorylation by overexpression of H3S57D causes slower growth and reduces 53BP1 chromatin binding upon induction of replication stress, while concomitant knockdown of RAD52 accelerates repair. 53BP1 controls repair of both stalled and collapsed forks and DSBs. Together with RIF1, 53BP1 prevents MRE11 nuclease-mediated reversed fork degradation, thus promoting fast fork restart. A competing pathway goes through the slower pathway of BIR, which involves BRCA1 and MUS-SLX complex-mediated cleavage[53,54]. At sites of DNA damage, 53BP1 forms a complex with RIF1 and Shieldin which limits excessive DSB end resection and orchestrates fill-in synthesis involving polymerase-α (polα)-primase[67,68]. Together with the fact that RAD52 is essential for SSA but not HR in human cells, our results suggest that the inability to phosphorylate H3S57 drives fork restart or DNA repair toward fast 53BP1-dependent pathways, and that H3S57D, which leads to lower 53BP1 binding, might redirect DNA repair towards SSA. Further studies will be required to elucidate how H3S57ph differentially affects distinct DNA repair pathways involved in replication stress responses. Phosphorylation of serine 139 on histone variant H2A.X, that occurs shortly after DNA damage, is mostly mediated by ATM and serves as a platform for recruiting proteins that mediate further histone modifications and DNA repair factors. This happens in human cells overexpressing both mutants of H3S57, suggesting that the initial steps of DNA damage sensing and signal transduction are H3S57ph-independent.

In our H3 mutant experiments in human cells, the modified histones are overexpressed, but the endogenous histones remain and so the phenotypes observed may not solely be due to deregulation of H3S57 modification. This is why it was important to study the roles of S57ph in fission yeast using gene replacements at the endogenous loci, where the S57 mutant histones constituted the only source of H3. In this context, we observed that the non-phosphorylatable S57A substitution completely abolished both HR and NHEJ, indicating that the phosphorylation of H3S57 may be essential for the early steps of both repair pathways. This could be for the general increase of accessibility of chromatin as well as for creating binding sites for specific DNA repair factors, for processing DNA ends and for the choice of a repair pathway. Surprisingly, fission yeast mutants with the constitutive phospho-mimicking H3S57D modification were also impaired in both HR and NHEJ repair pathways, albeit to a lesser extent than the A mutant. It is possible that the S57D mutant might have partial loss of function (in fission yeast) or a dominant-negative effect (in human cells overexpressing it), as it is not expected to fully recapitulate the biochemical effects of phosphorylation: the aspartic acid group provides only one additional negative charge, whereas phosphorylation adds two charges, thus increasing electrostatic repulsion between the DNA and the core histone and loosening the chromatin to a larger extent than the D or E mutation. Thus, it remains possible that H3S57ph is compatible with both NHEJ and HR. An alternative explanation for the compromised DNA repair pathways in fission yeast only expressing S57D is that the phosphorylation is likely dynamic whereas S57D is not. The inability to revert to an unphosphorylated state might prevent completion of repair, explaining the sensitivity of H3S57D mutants to replication stress and the formation of aberrant Rad52 foci in these cells.

Indeed, the importance of the dynamics of a histone PTM has been reported for H3K56 acetylation, whose establishment and removal are both required for its function in the DNA damage response. In yeast cells lacking both HST3 and HST4, histone deacetylases that reverse the acetylation of H3K56, the modification persists

at high levels throughout the entire cell cycle rather than being removed in G2/M and G1 phases. This results in high sensitivity of cells to genotoxic agents and temperature, along with reduced viability and replicative lifespan[69–72]. H3K56ac promotes HR and limits BIR[73,74]. However, despite the fact that the modification was discovered nearly two decades ago[23], the precise functions and mechanism of action of H3K56ac in DNA repair remain unclear. Given the similar biochemical effect obtained by both adding acetyl and phosphate groups to the histone (ref. [75] and this work), the rare occurrence of H3K56ac in human cells (which, on the contrary, is abundant in yeast[76]), as well as our finding that H3S57ph sequesters and inhibits acetylation of the neighbouring lysine, one might speculate that in human cells, H3S57ph has similar roles in DNA damage response to those of H3K56ac in yeast. Nevertheless, we might expect differences in the outputs of these modifications: acetylated histones are bound by bromodomain proteins, while phosphorylations are recognised by several different protein domains, including BRCT repeats that are found in diverse HR factors. Interestingly, we have identified several specific H3S57ph interactors that participate in various DNA damage response pathways, including DNA polymerase theta. This translesion polymerase is thought to regulate the balance between HR and alternative end joining (MMEJ)[4,77,78].

Our results also indicate that H3S57ph is not induced by DNA damage but is rather present on histones that are incorporated during DNA synthesis. Its function may therefore rely on exposure of the modification by removal of a binding factor. Such a mechanism has been reported for H4K20 di-methylation: this mark is bound by JMJD2A and L3MBTL1 proteins[79,80], and their removal upon DNA damage provides a binding site for 53BP1[20]. It will be interesting to address in future studies whether any of the H3S57ph readers antagonise the DNA damage responses and are actively removed to allow the process to proceed. Interestingly, BARD1 of the BRCA1 complex also binds H4K20, but in its unmodified form[21], exemplifying how DNA end processing and repair pathway choice may be fine-tuned by histone modifications. Similarly, H3S57 might serve as a binding site for different factors depending on its phosphorylation status. Future studies will be needed to investigate this possibility.

Finally, our whole-atom molecular dynamics simulations and genome-wide nucleosome-mapping experiments in budding yeast indicate that H3S57ph loosens histone H3-DNA contacts at the entry point to the nucleosome, rendering it more accessible. Nucleosome eviction is a key step of double strand DNA break repair that differentially affects different DNA repair pathways, potentially explaining the effects of preventing H3S57ph on DNA repair. Like H3K56ac, our data show that H3S57ph is a constitutive mark that is not induced upon DNA damage. Therefore, H3S57ph, similarly to H3K56ac, might affect multiple DNA metabolic processes in addition to DNA repair, such as facilitating nucleosome assembly[70,81] and promoting initiation and elongation steps of transcription[22,82]. Further studies will be required to address the roles of this modification in other DNA-dependent functions.

## Methods

### Antibodies

The anti-H3S57p antibody was produced by immunising two rabbits with the S57ph-containing peptide (IRRYQK{pSer}TELLIRKLPFQRLVR) (outsourced to Proteogenix). The collected sera were pooled and processed further to provide antibodies of highest specificity. Briefly, first, antibodies were affinity purified by bead-coupled S57ph peptide (as above) and depleted twice, first with the unmodified peptide (IRRYQKTELLIRKLPFQRLVR) and secondly with the N-tail peptide containing the S10ph that lies at a similar KST motif (QTAR(acet)-K{phSer}TGGKAPRKQL). The recovered purified antibody was diluted 1:1 with glycerol and stored at −20 °C. Specificity was evaluated with budding yeast expressing only H3S57A, dot blots with synthetic

peptides and peptide competitions, λ-phosphatase treatment as described in detail below. The antibody was used at 1:250 and 1:25 dilution for WB and IF, respectively.

The specific anti-H3K56ac antibody that we validated and used exclusively was from Cell Signalling (#4243). Other H3K56ac antibodies that we found to be less specific were: 39282 and 61062 from Active Motif, and 04-1135 from Millipore.

The list of other commercial antibodies used in human cells is provided in Supplementary Table 2.

The BG4 probe was purified from the pSANG10-3F-BG4 (#55756; Addgene[48]) using standard protocols[83]. Briefly, a clone of Rosetta bacteria transformed with the plasmid was cultured in 1 L LB+kanamycin medium. At 0.6 OD600, 1 M IPTG was added and grown at 28 °C to induce protein expression overnight (14 h). Expression was confirmed by SDS-PAGE. The bacterial pellet was resuspended in TES buffer (0.2 M Tris pH 8.0, 0.5 mM EDTA, 0.5 M sucrose) and incubated on ice with light shaking for one hour; 18 ml TES/4 (1/4 diluted TES) before incubation for an additional hour. The lysate containing the periplasmic material was cleared by centrifugation (8000 rpm for 30 min). TALON beads (2 ml) were pre-washed with 5x bead volume of PBS and incubated overnight with the protein. Beads were washed with 10x volume (PBS with 5 mM imidazole). BG4 was eluted with PBS + 500 mM imidazole. Imidazole was removed by successive rounds of dialysis in 3 L of PBS (3x1L, the second round was overnight). The purity and abundance of the purified probe was estimated by SDS-PAGE. Subsequently, the BG4 probe was diluted with 50% final glycerol and stored at −20 °C.

### Chemicals

The following products were from Sigma Aldrich unless otherwise stated: Thymidine (T1895), BrdU (10 mg/ml stock in H2O, freshly prepared, used at 30 μM; B5002), Hydroxyurea (HU, H8627), H2SO4 (320501), Butyric acid (1 M stock in DMSO; B5887), Trichostatin (TSA, 2 mg/ml – 6.6 mM stock in DMSO; T8552), Bleomycin (10 mg/ml stock in DMSO; B2434), Propionic anhydride (81942), Lactic acid, acetonitrile, ammonium bicarbonate, ICRF (2 mg/ml stock in DMSO, used at 2 μg/ml), Camptothecin (5 mM stock in DMSO, 1100/25, R&D Systems Europe), Etoposide (85 mM in DMSO stock; E1383), gelatin (G9391), FLAG immunoprecipitation kit (FLAGIPT1-1KT), acetyl-CoA (A2056-1mg). Micrococcal nuclease S7 (10107921001; Roche), stock 15,000 U ml⁻¹ in MNase buffer (see below). Chk1 inhibitors: AZD7762 (used at 50 or 500 nM), CHIR-124 (15 mM stock in DMSO, used at 1 μM), and SCH900776 (150 mM stock in DMSO, used at 10 μM) were from Seleckchem Euromedex (S1532, S2683 and S2735, respectively). Trypsin Gold MS grade (V5280; Promega), pyridostatin pentahydrochloride (4763; R&D Systems, used at 10 μM), recombinant full length H3 (C110A, 31207; Active Motif); highly active recombinant His-tagged CREB-binding Protein (mouse; CPB, CREBBP) was from AdipoGen (AG-40T-0016; A00269/C).

### Plasmids, peptides and RNAi

RNAi (siGenome SMARTpools; Dharmacon) for Chk1 (M-003255), Chk2 (M-003256), or siCtrl for luciferase, 5′-ACUGACGACUCUGCUA-CUC-3′ (Eurogentec, Seraing, Belgium). siRNA for RAD52 (3631456; Microsynth AG): 5′ GGAUGGUUCAUAUCAUGAATT.

Histone H3.1 plasmids used in this study were derivatives of the pDONR223-Hist1H3B clone (hORFeome v.3.1, gene ID 55755)[84]. In this vector, we have introduced silent mutations that initially were planned to confer resistance to a homemade shRNA (not used) while we introduced mutations to create S57D or S57A mutants (outsourced to Genscript). Each construct was subsequently cloned into pcDNA5-FRT-3xFLAG and then pQCXIP retroviral vector under a CMV promoter that was used to infect U2OS cells.

All peptides (both unlabelled or biotin-labelled) mentioned in the study were synthesised by Proteogenix.

## Cell culture

All cells were cultured in humidified incubator at 37 °C, 5% $CO_2$. Cells were washed in PBS and detached from plates with 0.05% trypsin solution supplemented with 0.5 mM EDTA. Cell confluency was at 70–80% or lower in all experiments. Normal human diploid foreskin fibroblasts (HDF) were from frozen stocks provided by J.Piette (CRBM, Montpellier) in 2001 and have been described previously[85,86]. U2OS were obtained from the ATCC. RPE, HeLa, MDA-MB-231, MCF10, PBMC, HEK293T, Jurkatt, KG1a, MV-4.11, CN-2, LNCaP were a gift from A. Castro lab (CRBM, Montpellier, France) and described in Vera et al., 2015[87]. The absence of mycoplasma contamination was confirmed by weekly testing in the IGMM facility (Mycoalert kit). U2OS, HEK293T, RPE, HDF and HeLa cells were cultured in DMEM Glutamax (Invitrogen) supplemented with 10% heat inactivated foetal bovine serum (FBS; Invitrogen) and 1x antibiotic (pen/strep) mixture (complete medium). All other cell lines were cultured in RPMI Glutamax supplemented with FBS and antibiotics as above. Mouse embryonic cells (mES) were cultured in ESGRO Clonal Grade Medium (SF-001-B supplemented with GSK3 beta inhibitor). Cells were collected by trypsinisation, washed with PBS, and cell pellets were flash frozen in liquid nitrogen before being stored at −80 °C. Cell synchronisation was achieved by a double thymidine block (DTB) or nocodazole. Briefly, cells were seeded in complete medium supplemented with 2 mM thymidine and incubated for 14–16 h, washed thoroughly with PBS and released in complete medium for 8–10 h. Then, thymidine was added again for 14–16 h to reach G1/S phase. Cells were washed well with PBS and fresh medium was added for synchronised S-phase progression and collected at indicated timepoints by trypsinisation. For arrest in M-phase, nocodazole (50 ng/µl) was added to asynchronous cultures for 16 h.

## siRNA

For RNAi, oligos were used at 40 nM final concentration and transfected with the calcium phosphate method (for CHK1 and CHK2), or with Lipofectamine™ RNAiMAX (ThermoFisher) transfection procedure (for RAD52; according to manufacturer's instructions). For complete Chk1 depletion, two rounds of depletion were performed in a window of 3 days when the cells were collected[88].

## FACS

For Fluorescence-activated cell sorting, cells were fixed in 70% EtOH, stained with propidium iodide, and analysed using FacsCalibur BD[89]. All FACS figures were prepared with Flowing Software (http://www.flowingsoftware.com/ - versions 2.4.1 and above), except Fig. 3b which was prepared with FlowJo. The gating strategy is presented in Supplementary Fig. 11.

## Flow cytometry-based analysis of chromatin-associated proteins

This analysis was performed as follows (see also[90]). Briefly, cells were collected by trypsinisation at 60–90% confluence ($10^5$ per sample), washed twice with PBS and centrifuged for 3 min at 400 $g$ at 4 °C, resuspended in 100 µl of extraction buffer (PBS/0.2% Triton X-100) and incubated on ice for 10 min (pre-extraction), followed by spin at 400 $g$ for 3 min at 4 °C. Cells were fixed/permeabilised in 100 µl of BD Cytofix/Cytosperm buffer (#554722), incubated for 20 min at RT. Subsequently, 0.5 ml of 1x Perm/Wash BD buffer (#554723) was added, cells were pipetted up and down and centrifuged for 3 min at 400 $g$. All centrifugations from now on were done at RT. If needed, cells were stored at this step at 4 °C for 2–3 days in storage buffer (PBS/3% FBS/0.09% sodium azide); before proceeding with staining, cells were washed in 0.5 ml of washing buffer. Otherwise, cells were processed directly for staining. 50 µl of washing buffer per sample with the appropriate concentration of the primary antibody (anti-RPA2 Abcam 1:200; anti-γH2A.X Cell Signalling 1:200; anti-BrdU BD 1:200; anti-53BP1 Bio-Techne 1:200) was used to resuspend cell pellet. Cells were

incubated for 1 h at RT, after which 0.5 ml of washing buffer was added and cells were centrifuged for 3 min at 400 $g$. Similarly, 50 µl of washing buffer was used per sample with the appropriate concentration of the secondary antibody (1:1000), and cells were incubated for 30 min at RT, followed by a wash in 0.5 ml of washing buffer. Cells were resuspended in 0.3–0.5 ml of analysis buffer (0.02% sodium azide, 250 µg/ml RNase, 0.5 µg/ml DAPI in PBS/BSA), incubated at 37 °C for 20–30 min, and either processed directly or stored in the dark at 4 °C. LSR Fortessa Becton Dickinson was used for cell analysis, run by FACSDiVa software (version 9.0.1). The gating strategy is provided in Supplementary Fig. 11.

## FLAG-tagged histone H3 expressing cells

U2OS cells stably expressing histone H3.1-3xFLAG constructs were obtained by viral transfections. Viral particles were created by mixing the pVSV-G, the pol/gag, and the pQCXIP vector (a gift from E. Julien, IRCM, Montpellier, France) expressing the histone allele and transfecting low confluency HEK293T cells by using the jetPEI method (Polyplus Transfection) to produce retroviral particles. After a second round of transfection 36 h after the first, fresh medium was added and left in culture for 48 h when the medium was collected, filtered through a 0.45 filter and stored aliquoted. Various dilutions of the viral particles were used to infect U2OS cells overnight. Selection with puromycin (2 µg/ml) was initiated 48 h post-infection until the control cells were dead. Clones were isolated by serial dilution.

## Subcellular fractionation

All buffers contained protease and phosphatase inhibitor cocktails (P8340 and P0044, respectively from Sigma). When necessary, 1 µg/ml TSA or 10 mM Butyric acid was added.

## Human cells: total cell extracts

Total cell lysates were obtained in RIPA buffer (50 mM Tris pH 8.0, 1% NP40, 0.5% Sodium deoxycholate, 0.1% SDS, 150 mM NaCl, 2 mM EDTA, 1 mM DTT). About $10^6$/100 µl were resuspended in RIPA, incubated on ice for 30 min and cleared by 30 min centrifugation at $16,000 \times g$.

## Histone purification

Acid-extracted histones were obtained using a standard protocol[91]. Briefly, nuclei were obtained with a hypotonic swelling and rotation for 30 min followed by a 12–16 h incubation in 0.2 M $H_2SO_4$ (Sigma). Proteins in solution were precipitated by TCA for at least an hour on ice, washed twice with ice-cold acetone, and the pellet was resuspended in water. Salt-extracted chromatin was prepared using a well described protocol[92].

## Purification of nuclei, chromatin, nucleoplasm

Nuclei, crude chromatin and nucleoplasm were prepared with the Mendez and Stillman method[93].

## Nucleosome fraction purification

Nucleosomal fractions were obtained with a sucrose gradient after MNase digestion according to the Ruthenburg lab protocol[94] with minor modifications[95]. Briefly, cells were suspended $6 \times 10^6$/ml in Buffer A (0.34 M Sucrose, 10 mM Hepes pH 7.9, 10 mM NaCl, 1.5 mM $MgCl_2$, 10% Glycerol, 5 mM DTT). An equal volume of Buffer A supplemented with 0.2% Triton-X100 was added (0.1% final) and left on ice for 10 min to lyse the cells. Nuclear pellet was obtained by centrifugation ($1300 \times g$, 5 min., 4 °C), resuspended in 3 ml Buffer A, layered on top of 7.5 ml sucrose cushion (30% sucrose, 10 mM Tris pH 7.5, 1.5 mM $MgCl_2$) and centrifuged ($10,000 \times g$, 20 min., in JS13.1 swing bucket rotor; Beckman) for additional purification. Nuclear pellet was resuspended in 550 µl MNase buffer (0.32 M sucrose, 50 mM Tris pH 7.5, 4 mM MgCl2, 1 mM $CaCl_2$). From that, 500 µl were incubated for

5 min in a water bath pre-warmed at 37 °C and 15 U of MNase was added for 30 min. The reaction was stopped by addition of EDTA (20 mM final) and incubation on ice for 10 min. Digested chromatin was recovered from the supernatant by centrifugation at 16,000 × $g$, 10 min at 4 °C. and was loaded onto a 5–29% sucrose gradient (in 10 mM sodium cacodylate, 1 mM EDTA, 0.5 mM EGTA, 50 mM KCl) in 13 × 51 mm round bottom Beckman tubes by using a Biocomp gradient master and centrifuged at 247,000 × $g$ for 3 h at 4 °C in a SW55Ti rotor. The fractionated nucleosomes were collected in fractions of 200 µl each from top to bottom with a pipette. From each fraction, 10 µl was used to extract DNA and confirm the nucleosomal profiles on 1.8% agarose gels after DNA clean-up. Pure mono- or di- or oligo- nucleosomal fractions were pooled and used for further experiments.

### Budding yeast

The YN1392 (K56A), YN2168 (S57A), YN2170 (S57E), YN2182 (K56AS57A), YN2186 (WT) *Saccharomyces* strains were a generous gift from Colin Logie[29] and were cultured in YEPD (1% yeast extract, 2% bacto-peptone, 2% dextrose). To obtain the acid-extracted histone-enriched fraction, we performed the following based on published protocols[96]. All buffers were supplemented with protease and phosphatase inhibitors. Yeast cell cultures were washed in water and the pellet was resuspended in 5 ml Buffer-1 (10 mM DTT, 100 mM PIPES pH 9.5) and shaken for 15 min at 30 °C. After centrifugation (3000 × $g$, 5 min, 4 °C), the pellet was resuspended in 5 ml Buffer-2 (1.2 M Sorbitol, 20 mM Hepes pH 7.9 supplemented with 0.8 mg/ml fresh zymolyase) and lightly shaken at 30 °C. Reactions were stopped when spheroplasted cells were obtained i.e. in about 50 min, when 10 ml cold Buffer-3 (1.2 M Sorbitol, 20 mM PIPES pH 6.8, 1 mM MgCl$_2$) was added and followed by centrifugation (3500 × $g$, 5 min, 4 °C). The cell pellet was resuspended in 5 ml cold NIB buffer (0.25 M sucrose, 60 mM KCl, 15 mM NaCl, 5 mM MgCl$_2$), incubated on ice for 20 min and recovered by centrifugation (3000 × $g$, 5 min, 4 °C). This step was repeated for a total of 3 washes. The pellet was further resuspended in Buffer-A (10 mM Tris pH 8.0, 0.5% NP-40, 75 mM NaCl), incubated on ice for 15 min and centrifuged (4000 × $g$, 5 min). This was followed by resuspension in Buffer-B (10 mM Tris pH 8.0, 0.4 M NaCl) and another centrifugation (4000 × $g$, 5 min). Histones were then enriched by H$_2$SO$_4$ (1 h incubation with rotation at 4 °C), precipitated by TCA/acetone and resuspended in water before being analysed in 18% SDS-PAGE.

### Nucleosome mapping.

1.2 × 10$^9$ yeast spheroplasts were digested with Micrococcal Nuclease (MNase), 2.4 units at 37 °C for 30 min with 3 mM CaCl$_2$. The reactions were stopped by addition of EDTA to a final concentration of 0.02 M and subsequently incubated with RNase A (0.1 mg) for 4 h at 37 °C, and treated with Proteinase K at 65 °C overnight. DNA was purified using phenol–chloroform extraction and concentrated by ethanol precipitation. The percentage of mononucleosomal DNA fragments was examined by 2% agarose gels. The integrity and size distribution of digested fragments were determined using the microfluidics-based platform Bioanalyzer (Agilent) prior to sample preparations and sequencing following Illumina standard protocol.

### Nucleosome calling.

MNase-seq paired-end reads were mapped to yeast genome (SacCer3, UCSC) using Bowtie[97] aligner and output files were imported in R[98]. Reads were trimmed to 50 bp maintaining the original centre and transformed to reads per million. Peak calling was performed, after noise filtering, with nucleR package[99] using the parameters: peak width = 147 bp, peak detection threshold = 35%, maximum overlap = 80 bp.

Nucleosome calls were considered as well-positioned when nucleR peak width score and height score were higher than 0.6 and 0.4, respectively, and as fuzzy otherwise.

## Fission yeast

### Strains and methods.

Standard media and methods were used[100,101]. All experiments were carried out in minimal medium with supplements (EMM6S) at 32 °C, except where otherwise noted. The *Schizosaccharomyces pombe* strains used in this study are shown in the strain table. To construct strains harbouring alterations in histone H3, the open reading frames of all three endogenous histone H3 loci (*hht1*, *hht2*, and *hht3*) were mutated so that the serine 57 residue was replaced by alanine (TCT to GCT) or aspartic acid (TCT to GAT). The mutation of each locus was achieved by replacement of the target histone H3 locus by the *ura4* gene in a strain harbouring *ura4-D18*, followed by integration of the mutated H3 coding region by homologous recombination and 5-FOA counterselection. Mutant alleles were then combined by genetic crosses and verified by Sanger sequencing.

For experiments in which cells were blocked in HU and then released to resume cell cycle progression, cells were grown to ~2 × 10$^7$ cells/ml in EMM6S at 32 °C and treated with 12 mM hydroxyurea (HU) for 4 h, resulting in S phase arrest. Cells were then washed using EMM6S by filtration and released to resume the cell cycle in EMM6S. For analysis of the sensitivities of mutant strains to DNA damaging agents, drop assays were performed. Cells were grown to 3–4 × 10$^7$ cells/ml in YE4S at 32 °C, sonicated using a BioRuptor (30 sec, 2 cycles, low setting; Diagenode, Liege, Belgium), and spotted on the indicated plates in 10-fold dilution series. Rich medium plates were used in this assay, with the following additions: 6 mM or 10 mM hydroxyurea (HU); 0.004% or 0.007% Methyl methanesulfonate (MMS), 6 µM Camptothecin (CPT). For UV exposure, plates were irradiated at 200 J/s. Unless otherwise indicated, all plates were incubated at 32 °C.

### Analysis of fission yeast cellular phenotypes.

For determination of cell size at division, cells were stained with 1 mg/ml Blankophor (MP Biomedicals), and cell length was determined using the Pointpicker plugin in Fiji (National Institutes of Health, USA). For DNA content analysis, cells were fixed in 70% cold ethanol, washed in 50 mM sodium citrate and treated with RNase A (0.1 mg/ml). Samples were then stained using 2 mg/ml propidium iodide, sonicated, and analysed using a BD Accuri C6 flow cytometer (BD Biosciences, Franklin Lakes, NJ, USA) and the Flowjo analysis software (FlowJo LLC, Ashland, OR, USA). Note that the fission yeast cell cycle has a short G1, and S phase occurs before cytokinesis. Cells therefore have a 2 C DNA content during most of their life cycle. A block in early S phase induced by exposure to hydroxyurea leads to cytokinesis taking place prior to DNA replication and produces a 1 C peak.

### Microscopy.

All microscopy experiments were performed on an inverted Zeiss Axio Observer (Carl Zeiss Inc.) equipped with a Plan-Apochromat 63X/1.4 NA immersion lens (Carl Zeiss Microscopy GmbH, Jena, Germany), a Lumencor Spectra X illumination system (Lumencor Inc., Beaverton, Oregon), a laser bench (Visitron Systems GmbH, Puchheim, Germany), and a Yokogawa CSU-W1 spinning disc confocal head (Yokagawa Life Science, Japan). Images were acquired using a Hamamatsu Orca Flash 4.0V2 sCMOS camera (Hamamatsu Photonics, Hamamatsu, Japan) and the Visiview software (Visitron Systems GmbH, Puchheim, Germany). Analyses were performed using Fiji (National Institutes of Health, USA).

### Detection of histone H3 S57 phosphorylation in fission yeast.

Fission yeast cells were grown to exponential phase in EMM6S at 32 °C. Cells were resuspended in SDS loading dye, boiled for 7 min, snap chilled in ice-cold water for 5 min, followed by addition of a protease and phosphatase inhibitor cocktail. Cells were broken using glass beads in a Precellys Evolution homogenizer (Bertin Technologies), and boiled again for 5 min. Extracts were then separated from the beads, snap chilled in ice-cold water, then centrifuged at 10,000 rcf for 10 min at 4 °C. Samples were subjected to SDS-PAGE and analysis by Western

blot. For phosphatase assays, treatment with lambda phosphatase (NEB) was performed after blotting. Membranes were incubated with 4000 U of lambda phosphatase for 3.5 h at 30 °C. For all analyses, membranes were blocked using 5% BSA in TBST. Antibodies were used at the following dilutions: affinity purified α-H3S57ph (this study), 1/500; α-histone H3 (Abcam 1791): 1/20000; α-phospho-Cdc2 (Cell Signalling 9111): 1/1000; α-rabbit HRP: 1/100000.

**Transformation assays for homologous recombination (HR) and non-homologous end joining (NHEJ).** Cells were grown in YE4S at 32 °C to $4-5 \times 10^6$ cells/ml and transformed using a standard lithium acetate transformation protocol[102]. 100 µL of cells at $10^9$ cells/ml were used for each transformation. The strains used for these assays were leucine auxotrophs, which allows for assessment of productive HR or NHEJ events. To assess HR, pJK148 (*leu1*) was digested using NruI and transformed into cells. Integration of the DNA fragment by HR at genomic *leu1* locus allows for growth on media lacking leucine. For NHEJ, pIRT2 (*S. cerevisiae* LEU2, *ars1*) was digested with EcoRV and transformed into cells. Re-ligation of the plasmid via NHEJ then allows for growth on media lacking leucine. As an internal control for transformation efficiency, pYL314 (*ars1*, natNT2), which provides resistance to nourseothricin, was constructed and included in each transformation. Transformations were first plated on YE4S and incubated overnight at 32 °C, then replica plated to YE4S-NAT and EMM4S-L plates.

The list of fission yeast strains used in this study is supplied as Supplementary Table 3.

## Xenopus

*Xenopus laevis* egg extract preparation, chromatin extraction and DNA replication assay have been previously described[89]. For the dephosphorylation experiment with λ-phosphatase, chromatin was extracted from an extract 120 min after addition of sperm. At the last step when Triton X100 is added, the sample was split into two tubes and centrifuged. Recovered chromatin pellets were resuspended in λ-phosphatase buffer (1x PMP buffer, 1 mM MnCl₂; P0753, New England Biolabs) treated with 10 U enzyme (or without) for 30 min at 30 °C. The reaction was stopped by boiling at 95 °C in Laemmli buffer and loaded onto 18% SDS-PAGE.

## Proteomics

For mass spectrometry analyses of *Xenopus* chromatin, replication reactions with a total of 500 µl extract were initiated by the addition of 1400 nuclei per µl, energy mix and cycloheximide, and were stopped at 15 or 60 min and chromatin was extracted. Pelleted chromatin was resuspended in 20 µl benzonase reaction buffer (50 mM Tris pH 8.8, 2 mM MgCl₂, 1 mM DTT supplemented with protease and phosphatase inhibitors) and treated with benzonase for 1 h on ice. Then, concentrated Laemmli buffer supplemented with DTT (10 mM final) was added and proteins were reduced at 37 °C for 30 min. Iodoacetamide (55 mM final) was added and left at ambient temperature in the dark for 30 min for alkylation. Samples were cleared by centrifugation and analysed with a 10% SDS-PAGE. After Coomassie stain and destain, each lane was sliced into 8 pieces, which were prepared for in-gel digestion, digested with trypsin, and peptides were excised and dried by vacuum centrifugation (see ref. 103). For phosphopeptide enrichment[104], homemade TiO₂ columns (600 µg and 250 µg) were packed with the help of isopropanol into gel loader tips plugged with a C8 piece at its narrow end. Each column was equilibrated with wash buffer (80% acetonitrile – ACN, 1% trifluoroacetic acid – TFA) and twice with loading buffer (wash buffer supplemented with 25% lactic acid).

Each peptide fraction was resuspended in 30 µl loading buffer and loaded onto the small column. With gentle pressure applied by a syringe (or 800–1800 rpm centrifugation steps), the sample was passed through the first column directly onto the big column and discarded. Each column was then washed once with loading buffer and twice with wash buffer. Sequential elution with increasing concentrations of NH₄OH (0.5% NH₄OH in H2O; 5% NH₄OH in H2O; 5% NH₄OH in H₂O and 20% ACN) permitted the recovery of phosphopeptides. Elutions from both columns were pooled, formic acid (33% final) was added and peptides were dried by vacuum centrifugation and stored at −20 °C.

## Histone peptide pull down assay

We used the protocol described in Wysocka (2006)[58]. All buffers contained protease, phosphatase inhibitors as well as 6.6 µM TSA, 10 mM Na(C₃H₇COO). More specifically, we used in total $5 \times 10^8$ asynchronous cells, which correspond to $8 \times 10^7$ cells per condition. The conditions were with biotin-labelled H3$_{51-72}$ synthetic peptide in either unmodified form, S57ph, K56ac, K56acS57ph doubly modified, a peptide with the amino acid sequence scrambled, or no peptide control. Magnetic T1 beads (2.5 ml in total; 0.4 ml per condition) were washed with 1.5 ml PBS thrice, and at the last wash were aliquoted into 6 tubes and PBS was removed. Two nmol of each peptide were added to 0.5 ml PBS and allowed to bind to the beads for 75 min with rotation at room temperature. This was followed by 3 washes in PBS supplemented with 0.1% BSA and stored at 4 °C until needed. All steps with cells were performed at 4 °C. Cells were lysed with hypotonic buffer A (10 mM Hepes pH 7.9, 10 mM KCl, 1.5 mM MgCl₂, 0.34 M sucrose, 10% glycerol, 1 mM DTT, 0.1% TX-100) for 10 min with rotation, and centrifuged at $4000 \times g$ for 4 min. Nuclear pellet was resuspended in 280 µl Buffer C (20 mM Hepes pH 7.9, 25% glycerol, 420 mM KCl, 1.5 mM MgCl₂, 0.2 mM EDTA, 1 mM DTT), incubated on a rotator for 30 min and then 500 µl of Buffer D (20 mM Hepes pH 7.9, 20% glycerol, 0.2 mM EDTA, 2 mM DTT) and Triton X-100 to a final concentration of 0.15% was added before brief sonication and centrifugation at 16,000 x g for 10 min. The supernatant was added to the saturated beads and incubated with rotation for 16 h. The beads were washed once with buffer D, twice with buffer D + 150 mM KCl, and twice with Buffer D + 300 mM KCl (1 ml each wash). Beads were further washed with PBS (0.5 ml) before elution with 100 µl 2× Laemmli buffer with 10 mM DTT for 15 min at 95 °C.

The 70% of the eluate was precipitated by methanol/chloroform method, resuspended in Laemmli buffer containing 10 mM DTT for reduction at 37 °C for 45 min, followed by alkylation with 55 mM iodoacetamide for 45 min. The entire sample was analysed by 4–20% SDS-PAGE. Each lane was excised into 9 bands and processed as above for in-gel trypsin digestion.

## Histone synthesis

The procedure used for the total chemical synthesis of H3 was essentially as previously described[33]. Briefly, H3 was assembled by native chemical ligation (NCL) of 3 fragments of H3 (aa Boc-H3(1-46)-Nbz, Boc-Thz-H3(47-89+S57ph)-Nle-Nbz, Boc-Cys(Trt)-H3(91-135)) via *tert*-butyloxycarbonyl (Boc) solid-phase peptide synthesis. M90 and M120 were subsituted to norleucine (Nle), which is structurally similar, to avoid Met oxidation during synthesis and handling as well as in biochemical studies, which would cause major structural perturbations. Fragments 2 and 3 were ligated via NCL in the presence of mercaptophenylacetic acid [MPAA] and 2-carboxy(ethyl phosphine) hydrochloride [TCEP HCl] in GnHCl buffer, pH 7.3, followed by conversion of the N-terminal thiazolidine of fragment 2 to the thioester using [Pd(alyll)Cl]₂. After NCL with fragment 1, the crude reaction mixture was dialysed and subjected to one-pot radical induced desulfurisation reaction to provide the full length protein H3-(1–135 ±S57ph), which was characterised by HPLC and mass-spectrometry.

## Kinome screening

The Kinome screen was performed by ProQinase (now Reaction Biology). The phosphorylation profile of the biotinylated sample peptide H3$_{51-72}$ at 1 µM was determined in a radiometric assay (33PanQinase® Activity Assay) on a panel of 190 Ser/Thr kinases (see Supplementary Data 2), using streptavidin-coated FlashPlate® HTS PLUS plates, as

follows: the reaction cocktails were pipetted into 96 well, V-shaped polypropylene microtiter plates ("assay plates") in the following order: 1) 10 µl of kinase solution, 2) 40 µl of buffer/ATP/ test sample mixture. The reaction cocktails contained 60 mM HEPES-NaOH, pH 7.5, 3 mM MgCl₂, 3 mM MnCl₂, 3 µM Na-orthovanadate, 1.2 mM DTT, 50 µg/ml PEG20000, 1 µM ATP/[γ-$^{33}$P]-ATP ($8.6 \times 10^{05}$ cpm per well), protein kinase (1–400 ng/50 µl) and sample peptide (1 µM). All PKC assays (except the PKC-mu and the PKC-nu assay) additionally contained 1 mM CaCl₂, 4 mM EDTA, 5 µg/ml Phosphatidylserine and 1 µg/ml 1.2-Dioleyl-glycerol. The MYLK2, CAMK1D, CAMK2A, CAMK2B, CAMK2D, CAMK4, CAMKK2 and DAPK2 assays additionally contained 1 µg/ml Calmodulin and 0.5 mM CaCl₂. The PRKG1 and PRKG2 assays additionally contained 1 µM cGMP. Each assay plate maximally comprised 95 kinase reaction cocktails, one well of each assay plate was used for a buffer/substrate control containing no enzyme. The assay plates were incubated at 30 °C for 60 min. Subsequently, the reaction cocktails were stopped with 20 µl of 4.7 M NaCl/35 mM EDTA. The reaction cocktails were transferred into 96-well streptavidin-coated FlashPlate® PLUS plates, followed by 30 min incubation at room temperature on a shaker to allow for binding of the biotinylated peptides to the streptavidin-coated plate surface. Subsequently, the plates were aspirated and washed three times with 250 µl of 0.9% NaCl. Incorporation of radioactive $^{33}$Pi was determined with a microplate scintillation counter (Microbeta, Perkin Elmer). Under the chosen conditions, proteins bind to the streptavidin-coated FlashPlate® PLUS plates only to very low extent. For evaluation of the results of the FlashPlate® PLUS-based assays, the background signal of each kinase (without substrate) had previously been determined in three independent experiments ($n = 3$) and mean background values (in cpm) had been calculated for each kinase at a radioactivity input of $8.0 \times 10^{05}$ cpm per well. The mean (±SD) of all kinase background values was $27 \pm 30$ cpm, and none of the background values exceeded 173 cpm. For each kinase, the activity value (raw counts of the kinase assay), the mean of two background values of the sample peptide and the corrected activity value (raw activity value minus sample peptide background) are compiled in Supplementary Data 2. Based on previous determinations of kinase background values in the FlashPlate® HTS PLUS-based assay, only corrected activity values of more than 116 cpm were considered as significant.

### Individual in vitro kinase and acetylation assays

For other kinase assays, commercial kinase CHK1 (0282-0000-1, Proqinase) or DYRK1A (a gift from Sandrine Ruchaud) were used in 20 µl reactions containing 50 µM ATP, γ-$^{33}$P-ATP, 1–2 µg of substrate, and 1× kinase buffer (50 mM Hepes pH 7.6, 10 mM MgCl, 1 mM DTT, 0.02% Triton X-100) at 30 °C for 20–30 min. For substrates, recombinant full length H3, H3$_{51-72}$ or H3$_{5-20}$ peptides were used; RBER-CHKtide (0581-0000-5; Proqinase) was used for positive control; reactions without substrate or reactions without kinase (to control for kinase autophosphorylation) were used for negative controls.

Reactions were stopped by pipetting them onto P81 phosphocellulose squares (20–134; Millipore). Unincorporated γ-$^{33}$P-ATP was washed away by 3 washes with 1% phosphoric acid and, after being dried, each filter was submerged in scintillation liquid (Ultima Gold, 6013329; Perkin) in appropriate vials and counted.

For acetylation assays, 0.25 µl of highly active recombinant His-tagged CREB-binding Protein (AdipoGen) was incubated with 4 µg synthetic full-length H3 (S57ph or unmodified), 30 µM Acetyl-CoA in HAT buffer containing 5% Glycerol, 50 mM NaCl, 50 mM Tris pH 8.0, 0.1 mM EDTA, 1 mM DTT. Reactions were performed in a total volume of 30 µl, at 30 °C, for 25 min, and stopped by the addition of 10 µl 4× Laemmli buffer.

### Chromatin immunoprecipitation (ChIP)

**Native ChIP.** N-ChIP was performed using a standard protocol[95] with the following minor modifications for U2OS cells. Native chromatin,

digested with MNase as described in the "nucleosome fraction purification" section above, was pre-cleared, SDS was added to 0.1% and incubated on ice for 12 min. Afterwards, IP buffer (20 mM Tris pH 8.1, 2 mM EDTA, 150 mM NaCl, 1% Triton X-100) with inhibitors was added to 1x concentration. We found that under these conditions we were able to increase the amount of H3S57ph immunoprecipitations, as with H3K79[105]. After storing 10% of the material for Input sample, aliquots were made and H3S57 antibody (8 µg), H3K9me3 (2 µg), H3 (2 µg), or control IgG (5 µg) were added to sample aliquots (each ChIP with approximately $10^7$ cells) and incubated for 14 h with rotation at 4 °C. Pre-washed beads slurry (20 µl per sample; Dynabeads protein A) pre-blocked in 5 mg/ml BSA in PBS was added and incubated for 2 h. The beads were washed with a total of 3 ml of each: wash A (50 mM Tris, 10 mM EDTA, 75 mM NaCl), wash B (as wash A but with 125 mM NaCl), and wash C (as wash A but with 175 mM NaCl).

For ChIP-WB, bound nucleosomes were eluted in 2x Laemmli buffer at 95 °C for 20 min with a short vortex every 3–4 minutes. Proteins were loaded on 18% SDS-PAGE and WB was performed using Clean-Blot IP Detection Reagent (1:250 in 2% BSA/TBS-T for 1 h) instead of the secondary antibodies mentioned above.

For ChIP-seq, bound nucleosomes were washed one last time with TE buffer containing 50 mM NaCl before being eluted with 300 µl elution buffer. RNAase was added to digest RNA for 2 h, and Proteinase K to digest proteins for another 2 h at 37 °C. DNA (from IP and Input samples) was precipitated and cleaned-up with phenol/chloroform/isomylalcohol and Qiagen clean-up columns. DNA concentration was estimated by Qubit fluorometry and libraries were prepared using the Illumina's TruSeq ChIP Sample Prep Kit (Low Throughput) according to the manufacture's protocol. ChIP libraries were sequenced on an Illumina HiSeq platform according to the manufacture's protocol. Sequence reads were mapped to the human genome (hg38) using bowtie2 with default settings. ChIP-Seq read enriched regions (peaks) were identified relative to control Input DNA using MACS2 software[106]. Two biological replicates were performed.

**Crosslinked ChIP.** X-ChIP was performed in U2OS cells using the protocol of Shah et al.[107]. Sonication was performed in the presence of 0.1% SDS in the lysis buffer and fragmentation (150–500 bp fragments) was verified with agarose gel electrophoresis. ChIP-WB was performed as described above.

### In vitro trypsin digestion of mononucleosomes

Mononucleosome fractions were prepared as described above, concentrated with Vivaspin 2 (10,000 MWCO; Sartorius) and the concentration was estimated by A280 spectrophotometry. About 1 µg was used in 20 µl reactions containing a range (0.5–10 ng) of trypsin concentrations (Gold MS grade; Promega) in the presence of 50 mM ammonium bicarbonate. Reactions took place at room temperature for 30–40 min and stopped with the addition of Laemmli buffer and boiled at 95 °C before being loaded on 18–20% SDS-PAGE. Once the trypsin:nucleosome ratio was determined (usually around 1:50 to 1:250 trypsin:nucleosome) for optimal tail clipping, we performed immunoblotting.

### Microscopy

Cells were seeded on 0.5–1% gelatin-coated glass coverslips and used in each experiment 48 h after seeding. After a wash with PBS, cells were fixed in 2–4% parafolmaldehyde in PBS for 15–20 min, then washed 3 times with PBS and subsequently permeabilised in PBS containing 2% BSA and 0.25% Triton X-100 for 20 min. Primary antibodies were diluted in the above buffer and incubated for 1 h at room temperature. After extensive washing with PBS + 0.25% Triton X-100, 30 min incubation with secondary antibodies (1:500–1000) diluted in the same buffer followed. After extensive washing, coverslips were mounted onto glass slides with 20 µl of ProLong Gold or ProLong Diamond

containing DAPI and left to dry overnight. Primary antibodies were used in the following dilutions: anti-H3S57ph (1:20–1:50), anti-γH2A.X (1:500), anti-FLAG (1:200–500), cyclin A (1:500).

The purified BG4 probe was used as follows (see also ref. 48). Briefly, BG4 was diluted 1:40 (around 100–200 μM) in 2% milk/PBS, and incubated for 45 min at 37 °C. This was followed by extensive washing and further incubation with the anti-FLAG, and then the secondary antibody as described above.

Images were acquired with upright Zeiss Axioimager Z1 microscope operated with the Metamorph 6.2.6 software (Molecular Devices). Confocal images were acquired with a Leica SP5 microscope. All settings were kept identical between the different conditions analysed.

Image analysis was performed in Fiji-ImageJ[108] using identical parameters. Foci counting and signal intensity were measured with in-house macros.

All graphs were prepared with GraphPad Prism v.5 or above, except Figs. 3a and 4a, which were made in Microsoft Excel. Tests for statistical significance are described in the figure legend of each figure panel.

## Proliferation assay
For proliferation assay, cells were seeded in 96 well plates at 500 cells per well, fixed in 4% paraformaldehyde and nuclei were stained with DAPI. Cell numbers were determined for each day by counting the stained nuclei using Cellomics ToxInsight imaging system (Thermo Scientific).

## Immunoblotting
SDS-PAGE for histone analyses was performed in 18% and for total lysates in 10% gels made in-house. For proteomics, pre-cast (Bio-Rad) gels were used. Proteins were transferred to PVDF membranes (IPVH00010; Millipore) and blocked with blocking buffer (2% BSA/TBS-T) for 1 h. Primary antibody solutions were made in blocking buffer and incubated for 12–16 h. Secondary antibodies, HRP-conjugated were incubated for 45–60 min. Antibody binding was detected with classical chemiluminescence (ECL West Pico or Femto; Pierce). We prepared separate gels for each individual histone antibody by loading in parallel identical amounts of sample. Loading and transfers onto PVDF membranes were confirmed by amidoblack staining for all membranes. Only one is presented here, but all are available upon request.

Peptide dot blots for antibody validation were performed as follows (see also ref. 109). Peptides were diluted in $H_2O$ and 2 μl of each were spotted on nictrocellulose membranes and left to dry before being blocked with 2% BSA/TBS-T for 1 h. For antibody competitions, peptides (50 μg in 3 ml of diluted antibody) were added to the antibody solutions and incubated for 1 h with rotation before incubation with the membrane overnight.

Protein quantification was performed with the BCA kit (Pierce).

## High performance liquid chromatography
Twenty micrograms of the acid-extracted nuclear fraction (histone-enriched) from U2OS cells was further fractionated by HPLC as in Drogaris et al. (2008)[39] with minor modifications. We used a longer column (Aeris WIDEPORE 3,6u XB-C18 250 × 2,1 mm, 00G-4482-AN, Phenomenex) and optimised the acetonitrile gradient (30–50% for 1 h) to better separate the histone H3 species. Individual fractions were evaporated to near dryness, re-suspended in Laemmli buffer, loaded on 18% gels for SDS-PAGE and analysed by immunoblotting.

## Purification of nascent DNA (iPOND)
To isolate proteins at the replication fork, mature chromatin or hydroxyurea-stalled forks, iPOND was performed in U2OS as follows (see also ref. 110). Briefly, U2OS cells were plated in 15 cm dishes and the day after $10^8$ cells (8–9 dishes) were used per condition. EdU pulse

was for 10 min, thymidine chase for 60 min, while HU 5 mM was added for 3 h. Cells were fixed, washed, stored, permeabilised, lysed, and click reaction was performed as in the original protocol. The nuclear pellet was resuspended in 2 ml lysis buffer 3 (1 mM EDTA, 0.5 mM EGTA, 10 mM Tris, pH 8.0, 100 mM NaCl, 0.1% sodium deoxycholate, 0.1% SDS). A sample without EdU and biotin azide was used as negative control while all buffers contained 1× protease and phosphatase inhibitors and all steps wedre performed on ice. Sonication was performed with a Covaris M220 sonicator in its dedicated 1 ml glass vials (each sample was sonicated in two and then pooled). Streptavidin agarose beads (100 μl per condition; 69203, Millipore) were washed with lysis buffer 3, blocked with 0.5% BSA/PBS for 2 h with rotation at room temperature, then washed twice with lysis buffer. Capture took place overnight with gentle rotation at 4 °C. This was followed by one wash in lysis buffer, one in wash buffer 1 (1% SDS, 50 mM Tris), two in wash buffer 2 (1% Triton X-100, 20 mM Tris, 2 mM EDTA, 500 mM NaCl), and two in wash buffer 1. Each wash was done with 1 ml, followed by short vortexing and centrifugation (1800 × $g$ for 1 min). Proteins were recovered from the beads with 100 μl 2× Laemmli buffer and boiling at 95 °C for 5 min. Input and capture fractions were loaded on gradient 18% gels and subjected to immunoblotting with H3 and S57ph antibodies. Using agarose beads and such stringent washes, we did not detect unspecific H3 binding in the No-Click sample hence we preferred it to magnetic beads.

## FLAG-Immunoprecipitation
U2OS cells were incubated in complete medium supplemented with 2 mM HU for 2 h, washed three times with complete medium, and released in complete medium for 1 h. Cells were harvested in cold PBS. Nuclear pellets were obtained as mentioned above and resuspended in lysis buffer (50 mM Tris HCl pH 7.4, 150 mM NaCl, 1 mM EDTA, 1% Triton-X100). Protein concentration was determined with the BCA kit (Pierce) and 5 mg of total protein were subjected to the FLAG immunoprecipitation protocol using the FLAGIPT1 Kit, according to manufacturer's instructions (Sigma). FLAG-tagged Histones were eluted from the beads using 0.5 mg/ml 3xFLAG peptide. Input and immunoprecipitated eluates were loaded on gradient 4–20% gels and subjected to immunoblotting with FLAG, Rad50 and RPA70 antibodies.

## Building of models and molecular dynamics simulations
The structure of the nucleosome particle was taken from the high-resolution crystal structure with PDB id 1KX5[111]. Cysteines were represented in their reduced form, while all histidines were protonated in ε. The initial structures for the wild-type histone tails and their phosphorylated version at the H3S57 position (on both H3) were prepared using VMD[112] based on human sequences from the UniProt Consortium[113], as reported elsewhere[114]. The models were then minimised in vacuo, neutralised (with 144 K+ ions), solvated (with explicit waters and 0.15 M of K+Cl−), and minimised in solution with positional restraints on the solute using our well-established multi-step protocol[115,116]. To produce the final models, the minimised structures were thermalised to 298 °C at NVT, and then simulated during 700 ns by means of Molecular Dynamics simulations at NPT (P = 1 atm). The first half of the simulations (350 ns) were considered as an equilibration step and were discarded for further analysis.

For representing the histone proteins, we used the ff12SB force field[117], which is compatible with the phosphoserine parameters available in the literature[118,119]. The DNA was represented using parmbsc1[120], and the entire system was surrounded by a truncated octahedral box of ~47,000 TIP3P water molecules[121], applying Dang's parameters on ions[122]. Ions were initially placed randomly, at a minimum distance of 5 Å from the solute and 3.5 Å from one another. All systems were simulated using the Berendsen algorithm[123] to control the temperature and the pressure, with a coupling constant of 5 ps. Centre of mass motion was removed every 10 ps to limit build-up of the translational

kinetic energy of the solute. SHAKE[124] was used to keep all bonds involving hydrogen at their equilibrium values, in addition to the hydrogen mass repartitioning approach[125], which allowed us to use a 4 fs step for the integration of Newton equations of motion. Long-range electrostatic interactions were accounted for by using the Particle Mesh Ewald method[126] with standard defaults and a real-space cut-off of 9 Å. All simulations were carried out using the PMEMD CUDA code module[127] of AMBER 14[128], and analysed with CPPTRAJ[129], VMD[112], and CURVES+[130]. Supplementary Data 4 and 5 are.pdb files allowing visualisation of representative structures of unmodified nucleosomes and nucleosomes containing H3S57ph, respectively.

## Reporting summary
Further information on research design is available in the Nature Portfolio Reporting Summary linked to this article.

## Data availability
The proteomics data generated in this study have been deposited in the ProteomeXchange Consortium via the PRIDE partner repository[131] with the dataset identifier PXD011328 and 10.6019/PXD011328 (https://www.ebi.ac.uk/pride/). The MNase-seq data Array Express under the accession number E-MTAB-6901. H3S57ph ChIP-seq data have been deposited in the GEO database, under the accession number GSE119742. MD trajectories are publicly available through our BigNasim database (http://mmb.irbbarcelona.org/BIGNASim/). Source data are provided with this paper.

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

## Acknowledgements

This study was undertaken with funding from the ANR (ANR-09-BLAN-0252) and Ligue Nationale Contre le Cancer, "Equipe labellisée," grants EL2010.LNCC/DF, EL2013.LNCC/DF and EL2018.LNCC/DF; Inserm/ITMO cancer "epigenetics and cancer" 2015 grant. N.P. was further supported by a UICC Yamagiwa-Yoshida Memorial International Cancer Study Grant and the Labex Who am I?. P.D.D. is a PEDECIBA (Programa de Desarrollo de las Ciencias Básicas) and SNI (Sistema Nacional de Investigadores, Agencia Nacional de Investigación e Innovación, Uruguay) researcher. M.O. is an ICREA (Institució Catalana de Recerca i Estudis Avançats) academia researcher. Spanish Ministry of Science [BFU2014-61670-EXP, BFU2014-52864-R to M.O.]; Catalan SGR, the Instituto Nacional de Bioinformática; European Research Council (ERC SimDNA to M.O); European Union's Horizon 2020 research and innovation program [676556 to M.O.]; Biomolecular and Bioinformatics Resources Platform (ISCIII PT 13/0001/0030 to M.O.) co-funded by the Fondo Europeo de Desarrollo Regional (FEDER); MINECO Severo Ochoa Award of Excellence (Government of Spain, awarded to IRB Barcelona). P.-Y.J.W. was supported by grant EPIG201504 (Inserm/ITMO Cancer, Plan Cancer 2014-2019; funding for D.M.-S.) and the Institut National du Cancer (INCA, PLBIO 2015-117; funding for B.S.). P.-Y.J.W. and D.C. were supported by the Région Nouvelle Aquitaine (program CHESS, grant agreements 15963520 and 15964420). D.C. also acknowledges funding from the Région Bretagne (program SAD). A.B. holds the Jordan and Irene Tark Academic Chair. The project used computer resources from the Barcelona Supercomputing Centre (BSC). We acknowledge the Montpellier Ressources Imagerie facility (BioCampus Montpellier, Centre National de la Recherche Scientifique (CNRS), INSERM, University of Montpellier), a member of the national infrastructure France BioImaging supported by the French National Research Agency (ANR-10-INBS-04, "Investments for the future"). We thank C. Logie for yeast H3 mutants; K. Hached, J. Vera, A. Castro, D.C. Allis, E. Julien, M. Mechali, B. Pradet-Balade, S. Ruchaud for materials.

## Author contributions

N.P., P.-Y.J.W. and L.K. designed, supervised and performed experiments, analysed data and wrote the manuscript; P.D.P. and M.O. performed molecular dynamics analysis and analysed data; B.S., D.S.-T., D.M.-S., I.B.H, D.B., R.L. and M.G. performed experiments and analysed data; D.H., T.N.A., N.T., S.G., K.D. and C.Z. performed experiments; R.F. provided supervision to M.G. and contributed ideas; M.J, S.K.M. and A.B. synthesised full-length histone H3 bearing or not S57ph; D.C. developed methodology used; V.R. acquired data; M-N.P. provided supervision; D.F. conceived, designed and supervised the study, analysed data, wrote the manuscript, and provided the lab environment for the study.

## Competing interests

The authors declare no competing interests.
