## [Peer Review File · Nature Communications]

Histone H3 serine-57 is a CHK1 substrate whose phosphorylation affects DNA repairEditorial Note: This manuscript has been previously reviewed at another journal that is not operating a transparent peer review scheme. This document only contains reviewer comments and rebuttal letters for versions considered at *Nature Communications*.

Reviewers' comments:

Reviewer #1 (Remarks to the Author):

This manuscript studies the role of H3S57 phosphorylation in genome stability. Although there is no doubt that H3 is phosphorylated in S57 in both mammalian cells and in yeast as revealed by phosphoproteomics or using a specific antibody, the revised manuscript fails to convince about the importance of the mark in replication stress and DNA repair pathways choice. The modification is not convincingly induced upon DNA damage and it is not clear whether it is specific to a certain type of damage and why. Moreover, the experiments to demonstrate DNA repair pathway choice are indirect using co-depletion experiments for RAD52 and assessing γ -H2AX.

Overall, this manuscript contains different interesting findings but they are not well connected to each other to a solid story.

Reviewer #2 (Remarks to the Author):

I have assessed the manuscript and the responses to the referees. In my opinion, the work is novel and interesting enough so that it deserves publication. I do agree with the reviewers in that some of the data are over-interpreted, but in any case, in my view the authors have worked significantly to address the initial comments and this should suffice to communicate a novel discovery.

In my own view, the only real question here is whether this histone modification is only related to DNA replication and repair, or, more likely, most DNA transactions will be affected by it. However, this can be addressed in subsequent works and should not delay the publication of this work.

Reviewers' comments:

Reviewer #1 (Remarks to the Author):

This manuscript studies the role of H3S57 phosphorylation in genome stability.

This is a misrepresentation of our study: our manuscript does not just “study the role of H3S57ph in genome stability”. Below is a summary of our major findings:

- 1) We identify the novel H3S57ph mark using mass spectrometry and unique tools including highly specific antibodies and fully chemically synthesised histone H3 in both phosphorylated and unphosphorylated forms.
- 2) We show that this modification is conserved in mammals, other vertebrates and two distantly related types of yeast.
- 3) We identify Chk1 as the major kinase responsible for this modification in mammalian cells.
- 4) We show that, like H3K56ac in yeast, H3S57ph is incorporated into chromatin during replication and is absent in mitosis.
- 5) We demonstrate that in both fission yeast and mammalian cells, alteration of H3S57ph sensitises cells to replication stress, as is also the case for H3K56ac.
- 6) We reveal that H3S57ph is involved in DNA repair: the modification counteracts 53BP1 foci formation in mammalian cells and is required for both NHEJ and HR in fission yeast.
- 7) We find that H3S57ph hinders H3K56 acetylation, potentially explaining the low abundance of K56ac in mammalian cells. We further provide whole atom molecular dynamics modeling data for the entire nucleosome that suggests that this modification loosens the nucleosome, with implications for its impact on chromatin structure.

Our manuscript thus begins with the identification of a novel mark, proceeds with the demonstration of its conservation throughout eukaryotes, elucidates its function in a key cellular process, and concludes with structural insight into the molecular mechanisms underlying its function. The link between H3S57ph and genome instability thus represents only a small portion of our work, and limiting it to this unjustly diminishes the breadth, depth, and importance of our study.

Although there is no doubt that H3 is phosphorylated in S57 in both mammalian cells and in yeast as revealed by phosphoproteomics or using a specific antibody, the revised manuscript fails to convince about the importance of the mark in replication stress and DNA repair pathways choice.

While trying to understand this blanket statement, which was neither shared by reviewer 2 nor substantiated by specific critiques of our assays or interpretations, we realised that we may have inadvertently induced some confusion by stating in the abstract that H3S57ph influences DNA repair pathway choice in both fungi and mammalian cells. We appreciate that this may have been taken to mean that H3S57ph promotes one pathway at the expense of another. While this may yet be the case, as suggested by our data in mammalian cells, we are aware that this is not proven by our findings, and both NHEJ and HR are abrogated in fission yeast expressing only H3S57A alleles. We recognise that our wording may have influenced the reviewer's perception of our study, which would be unfortunate. Our data nevertheless show unambiguously that the mark is important for replication stress responses and DNA repair in both mammalian cells and in fission yeast. We have now also altered the text to remove any possible confusion.

The modification is not convincingly induced upon DNA damage and it is not clear whether it is specific to a certain type of damage and why.

First, the reviewer appears to have misunderstood the results of our experiments: in our revised manuscript, we show that H3K57ph levels are NOT increased upon DNA damage. This is similar to what was reported for H3K56ac, a modification that has been shown by different labs to be involved in DNA repair.

Second, our results clearly demonstrate that impairing H3S57ph dynamics affects the cellular response to specific types of DNA damage. Indeed, fission yeast cells harboring H3S57A and H3S57D are sensitive to different replication stresses leading to DNA damage (MMS, HU, and CPT) but not to UV-induced DNA damage (Figure 5).

Third, we are astonished that the reviewer seems to feel that we need to resolve all of the biology of this histone mark in a single paper. We note that in the case of the modification of the adjacent residue, H3K56ac, what the

reviewer asks for was addressed by a large body of work published in high-impact journals over two decades - with still many outstanding questions - and not in a single publication (see abridged list below).

The H3K56ac mark was described in yeast (Ozdemir et al., J Biol Chem, 2005; Hyland et al., Mol Cell Biol., 2005; Recht et al., PNAS, 2005) and shown to be involved in transcription (Xu et al., Cell, 2005) and replication stress responses, possibly by loosening nucleosomes (Masumoto et al., Nature, 2005). The enzyme that carries out this modification in yeast was described in two papers, which also showed its involvement in responses to various types of DNA damage (Han et al. Science, 2007; Driscoll et al. Science, 2007). The roles of H3K56ac in preserving genome integrity were confirmed in several other papers (Celic et al. Curr Biol 2006; Simoneau, et al, Genetics, 2015). In yeast, this mark was found to be involved in replication-coupled nucleosome assembly (Li et al., Cell, 2008), double strand break repair (without specifying which pathway, Chen et al., Cell, 2008), NHEJ (Miller et al., Nat Struct Mol Biol, 2010), sister-chromatid recombination (Munoz-Galvan et al., PLoS Genet, 2013), mismatch repair (Kadyrova et al., PLoS Genet, 2013), Break-Induced Replication (Che et al., PLoS Genet 2015) and UV-responsive nucleotide excision repair (Khan and Chaudhuri, DNA Repair, 2022).

This non-exhaustive list encompasses 16 publications spanning 17 years. In this regard, expecting a single histone mark to be specific to a certain type of DNA repair, or indeed for a single paper to demonstrate a definitive mechanism of action of a histone mark, reveals a biased appreciation of the literature in this field.

Moreover, the experiments to demonstrate DNA repair pathway choice are indirect using co-depletion experiments for RAD52 and assessing g-H2AX.

This comment disregards a large part of the experiments that we performed to assess DNA repair pathway choice, which in fact include direct assays in fission yeast.

In mammalian cells, there is no way to replace all copies of histone H3 with mutant versions, as the reviewer is aware; the best we can do is interfere with endogenous histones by expressing ectopic mutant versions. Nevertheless, using such an approach combined with state-of-the-art single-cell FACS assays (Figure 4), we showed that H3S57ph is involved in replication stress and interplays with 53BP1, which promotes NHEJ and is involved in stalled replication fork restart as well, and RAD52, which is involved in classical HR and SSA.

To test directly whether H3S57ph is required for particular DNA repair pathways, we used fission yeast strains that express only non-phosphorylatable (H3S57A) or phosphomimetic (H3S57D) versions of the protein. These alleles were integrated at all three endogenous histone H3 loci and expressed at endogenous levels. We then performed direct DNA repair assays for both NHEJ and HR and showed that the modification is important for both pathways (Figure 5E). The failure of the reviewer to acknowledge these substantial data sets questions the fairness of his/her review.

Overall, this manuscript contains different interesting findings but they are not well connected to each other to a solid story.

We are surprised by this viewpoint, which suggests an apparent preference for oversimplification; we feel that such attitudes are damaging for science. As summarised in our response above, our study begins with the identification of a new and conserved histone mark, proceeds to identify its writer and unravel its roles in cellular processes, and concludes with a likely molecular mechanism by which this mark carries out its functions. Our conclusions could only be reached by multiple orthogonal lines of experimentation. Importantly, our story reflects the complexities of the biology of this histone mark, which, as pointed out by reviewer 2, is likely to be involved not only in DNA replication/repair but also in “most DNA transactions.”

Reviewer #2 (Remarks to the Author):

I have assessed the manuscript and the responses to the referees. In my opinion, the work is novel and interesting enough so that it deserves publication.

We appreciate this positive assessment of our study and the clear recommendation for publication.

I do agree with the reviewers in that some of the data are over-interpreted, but in any case, in my view the authors have worked significantly to address the initial comments and this should suffice to communicate a novel discovery.

We would like to emphasise the fact that our original manuscript was submitted not to *Nature Communications* but transferred from *Nature* to *Nature Structural and Molecular Biology* (in the same short format, which was not the ideal format for the transferred paper). We also understand from the original reviewers' opinion that it was not strong enough to be published as it was, and we have taken into account the original referees' criticisms with more single-cell work and the genetic fission yeast model. We thank the reviewer for recognising the

improvements to our original manuscript and for the endorsement to communicate our discovery to the scientific community in a general journal with broad impact.

In my own view, the only real question here is whether this histone modification is only related to DNA replication and repair, or, more likely, most DNA transactions will be affected by it. However, this can be addressed in subsequent works and should not delay the publication of this work.

We agree that H3S57ph may indeed have additional functions in other DNA-related processes and have now mentioned this in the discussion. We also agree with the reviewer's recommendation that this can be addressed in subsequent work. The publication of our study would indeed open exciting new areas of investigation and allow the scientific community to build upon our findings.